# Poincaré Recurrence, Cycles and Spurious Equilibria in Gradient-Descent-Ascent for Non-Convex Non-Concave Zero-Sum Games

**Lampros Flokas**[*]
Department of Computer Science
Columbia University
New York, NY 10025
lamflokas@cs.columbia.edu

**Emmanouil V. Vlatakis-Gkaragkounis**[*]
Department of Computer Science
Columbia University
New York, NY 10025
emvlatakis@cs.columbia.edu

**Georgios Piliouras**
Engineering Systems and Design
Singapore University of Technology and Design
Singapore
georgios@sutd.edu.sg

## Abstract

We study a wide class of non-convex non-concave min-max games that generalizes over standard bilinear zero-sum games. In this class, players control the inputs of a smooth function whose output is being applied to a bilinear zero-sum game. This class of games is motivated by the indirect nature of the competition in Generative Adversarial Networks, where players control the parameters of a neural network while the actual competition happens between the distributions that the generator and discriminator capture. We establish theoretically, that depending on the specific instance of the problem gradient-descent-ascent dynamics can exhibit a variety of behaviors antithetical to convergence to the game theoretically meaningful min-max solution. Specifically, different forms of recurrent behavior (including periodicity and Poincaré recurrence) are possible as well as convergence to spurious (non-min-max) equilibria for a positive measure of initial conditions. At the technical level, our analysis combines tools from optimization theory, game theory and dynamical systems.

## 1 Introduction

Min-max optimization is a problem of interest in several communities including Optimization, Game Theory and Machine Learning. In its most general form, given an objective function $r : \mathbb{R}^n \times \mathbb{R}^m \to \mathbb{R}$ and we would like to solve the following problem

$$(\boldsymbol{\theta}^*, \boldsymbol{\phi}^*) = \underset{\boldsymbol{\theta} \in \mathbb{R}^n}{\arg \min} \, \underset{\boldsymbol{\phi} \in \mathbb{R}^m}{\arg \max} \, r(\boldsymbol{\theta}, \boldsymbol{\phi}). \tag{1}$$

This problem is much more complicated compared to classical minimization problems, as even understanding under which conditions such a solution is meaning-full is far from trivial [DP18, MPR+17, OSG+18, JNJ19]. What is even more demanding is understanding what kind of algorithms/dynamics are able to solve this problem when a solution is well defined.

---

[*]Equal contribution

Recently this problem has attracted renewed interest motivated by the advent of Generative Adversarial Networks (GANs) and their numerous applications [GPM+14, RMC16, IZZE17, GPM+14, ZXL17, ACB17, LTH+17, SGZ+16]. A classical GAN architecture mainly revolves around the competition between two players, the generator and the discriminator. On the one hand, the generator aims to train a neural network based generative model that can generate high fidelity samples from a target distribution. On the other hand, the discriminator's goal is to train a neural network classifier than can distinguish between the samples of the target distribution and artificially generated samples. While one could consider each of the tasks in isolation, it is the competitive interaction between the generator and the discriminator that has lead to the resounding success of GANs. It is the "criticism" from a powerful discriminator that pushes the generator to capture the target distribution more accurately and it is the access to high fidelity artificial samples from a good generator that gives rise to better discriminators. Machine Learning researchers and practitioners have tried to formalize this competition using the min-max optimization framework mentioned above with great success [AGL+17, Ma18, GXC+18, YFW+19].

One of the main limitations of this framework however is that to this day efficiently training GANs can be a notoriously difficult task [SGZ+16, MPPS17, MPP18, KAHK17]. Addressing this limitation has been the object of interest for a long line work in the recent years [MGN18, MPPS17, PV16, RMC16, TGB+17, BSM17, GAA+17]. Despite the intensified study, very little is known about efficiently solving general min-max optimization problems. Even for the relatively simple case of bilinear games, the little results that are known have usually a negative flavour. For example, the continuous time analogue of standard game dynamics such as gradient-descent-ascent or multiplicative weights lead to cyclic or recurrent behavior [PS14, MPP18] whereas when they are actually run in discrete-time[2] they lead to divergence and chaos [BP18, CP19, BP19a]. While positive results for the case of bilinear games exist, like extra-gradient (optimistic) training ([DISZ18, MLZ+19a, DP19]) and other techniques [BRM+18, GHP+19a, GBV+19, ALW19], these results fail to generalize to complex non-convex non-concave settings [OSG+18, LLRY18, SRL18]. In fact, for the case of non-convex-concave optimization, game theoretic interpretations of equilibria might not even be meaningful [MR18, JNJ19, ADLH19].

In order to shed some light to this intellectually challenging problem, we propose a quite general class of min-max optimization problems that includes bilinear games as well as a wide range of non-convex non-concave games. In this class of problems, each player submits its own decision vector just like in general min-max optimization problems. Then each decision vector is processed separately by a (potentially different) smooth function. Each player finally gets rewarded by plugging in the processed decision vectors to a simple bilinear game. More concretely, there are functions $F : \mathbb{R}^n \to \mathbb{R}^N$ and $G : \mathbb{R}^m \to \mathbb{R}^M$ and a matrix $U_{N \times M}$ such that

$$r(\boldsymbol{\theta}, \boldsymbol{\phi}) = \boldsymbol{F}(\boldsymbol{\theta})^\top U \boldsymbol{G}(\boldsymbol{\phi}). \tag{2}$$

We call the resulting class of problems Hidden Bilinear Games.

The motivation behind the proposed class of gamess is actually the setting of training GANs itself. During the training process of GANs, the discriminator and the generator "submit" the parameters of their corresponding neural network architectures, denoted as $\boldsymbol{\theta}$ and $\boldsymbol{\phi}$ in our problem formulation. However, deep networks introduce nonlinearities in mapping their parameters to their output space which we capture through the non-convex functions $F, G$. Thus, even though hidden bilinear games do not demonstrate the full complexity of modern GAN architectures and training, they manage to capture two of its most pervasive properties: *i) the indirect competition of the generator and the discriminator* and *ii) the non-convex non-concave nature of training GANs*. Both features are markedly missing from simple bilinear games.

**Our results.** We provide, the first to our own knowledge, global analysis of gradient-descent-ascent for a class of non-convex non-concave zero-sum games that by design includes both features of bilinear zero-sum games as well as of single-agent non-convex optimization. Our analysis focuses on the (smoother) continuous time dynamics (Section 4,5) but we also discuss the implications for discrete time (Section 7). The unified thread of our results is that gradient-descent-ascent can exhibit a variety of behaviors antithetical to convergence to the min-max solution. In fact, convergence to a set of parameters that implement the desired min-max solution (as e.g. GANs require), if it actually

happens, is more of an accident due to fortuitous system initialization rather than an implication of the adversarial network architecture.

Informally, we prove that these dynamics exhibit conservation laws, akin to energy conservation in physics. Thus, in contrast to them making progress over time their natural tendencies is to "cycle" through their parameter space. If the hidden bilinear game $U$ is 2x2 (e.g. Matching Pennies) with an interior Nash equilibrium, then the behavior is typically periodic (Theorem 3). If it is a higher dimensional game (e.g. akin to Rock-Paper-Scissors) then even more complex behavior is possible. Specifically, the system is formally analogous to Poincaré recurrent systems (e.g. many body problem in physics) (Theorems 6, 7). Due to the non-convexity of the operators $F, G$, the system can actually sometimes get stuck at equilibria, however, these fixed points may be merely artifacts of the nonlinearities of $F, G$ instead of meaningful solutions to the underline minmax problem $U$. (Theorem 8).

In Section 7, we show that moving from continuous to discrete time, only enhances the disequilibrium properties of the dynamics. Specifically, instead of energy conservation now energy increases over time leading away from equilibrium (Theorem 9), whilst spurious (non-minmax) equilibria are still an issue (Theorem 10). Despite these negative results, there are some positive news, as at least in some cases we can show that time-averaging over these non-equilibrium trajectories (or equivalently choosing a distribution of parameters instead of a single set of parameters) can recover the min-max equilibrium (Theorem 4). Technically our results combine tools from dynamical systems (e.g. Poincaré recurrence theorem, Poincaré-Bendixson theorem, Liouville's theorem) along with tools from game theory and non-convex optimization.

Understanding the intricacies of GAN training requires broadening our vocabulary and horizons in terms of what type of long term behaviors are possible and developing new techniques that can hopefully counter them.

The structure of the rest of the paper is as follows. In Section 2 we will present key results from prior work on the problem of min-max optimization. In Section 3 we will present the main mathematical tools for our analysis. Sections 4 through 6 will be devoted to studying interesting special cases of hidden bilinear games. Section 8 will be the conclusion of our work.

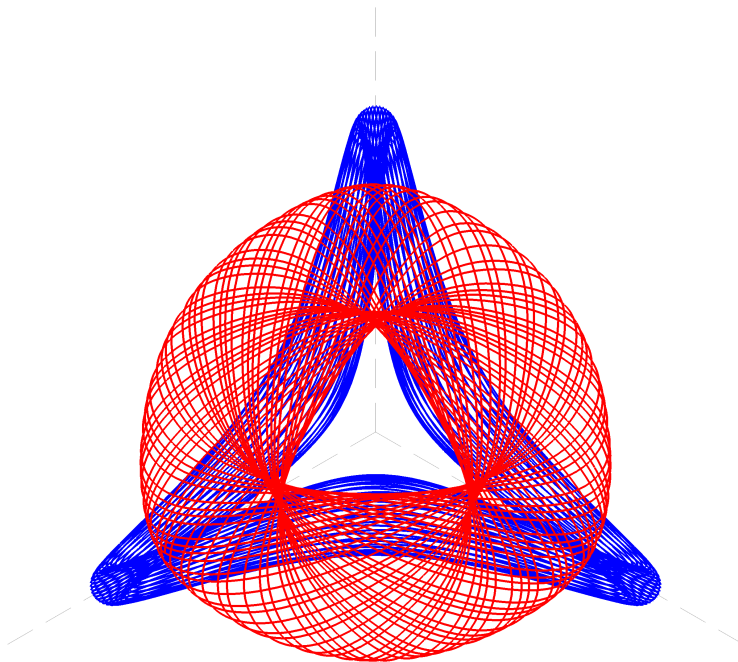

Figure 1: Trajectories of a single player using gradient-descent-ascent dynamics for a hidden Rock-Paper-Scissors game with sigmoid activations. The different colors correspond to different initializations of the dynamics. The trajectories exhibit Poincaré recurrence as expected by Theorem 7.

## 2 Related Work

*Non-equilibrating dynamics in game theory.* [KLPT11] established non-convergence for a continuous-time variant of Multiplicative Weights Update (MWU), known as the replicator dynamic, for a 2x2x2 game and showed that as a result the system converges to states whose social welfare dominates that of all Nash equilibria. [PPP17] proved the existence of Li-Yorke chaos in MWU dynamics of 2x2 potential games. From the perspective of evolutionary game theory, which typically studies continuous time dynamics, numerous nonconvergence results are known but again typically for small games, e.g., [San10]. [PS14] shows that replicator dynamics exhibit a specific type of near periodic behavior in bilinear (network) zero-sum games, which is known as Poincaré recurrence. Recently, [MPP18] generalized these results to more general continuous time variants of FTRL dynamics (e.g. gradient-descent-ascent). Cycles arise also in evolutionary team competition [PS18] as well as in network competition [NMP18]. Technically, [PS18] is the closest paper to our own as it studies evolutionary competition between Boolean functions, however, the dynamics in the two models are different and that paper is strictly focused on periodic systems. The papers in the category of cyclic/recurrent dynamics combine delicate arguments such as volume preservation and the existence of constants of motions ("energy preservation"). In this paper we provide a wide generalization of these type of results by establishing cycles and recurrence type of behavior for a large class of non-convex non-concave games. In the case of discrete time dynamics, such as standard gradient-descent-ascent, the system trajectories are first order approximations of the above motion and these conservation arguments do not hold exactly. Instead, even in bilinear games, the "energy" slowly increases over time [BP18] implying chaotic divergence away from equilibrium [CP19]. We extend such energy increase results to non-linear settings.

*Learning in zero-sum games and connections to GANs.* Several recent papers have shown positive results about convergence to equilibria in (mostly bilinear) zero-sum games for suitable adapted variants of first-order methods and then apply these techniques to Generative Adversarial Networks (GANs) showing improved performance (e.g. [DISZ18, DP19]). [BRM$^+$18] made use of conservation laws of learning dynamics in zero-sum games (e.g. [BP19b]) to develop new algorithms for training GANs that add a new component to the vector field that aims at minimizing this energy function. Different energy shrinking techniques for convergence in GANs (non-convex saddle point problems) exploit connections to variational inequalities and employ mirror descent techniques with an extra gradient step [GBVL18, MLZ$^+$19a]. Moreover, adding negative momentum can help with stability in zero-sum games [GHP$^+$19b]. Game theoretic inspired methods such as time-averaging work well in practice for a wide range of architectures [YFW$^+$19].

## 3 Preliminaries

### 3.1 Notation

Vectors are denoted in boldface $\boldsymbol{x}, \boldsymbol{y}$ unless otherwise indicated are considered as column vectors. We use $\|\cdot\|$ corresponds to denote the $\ell_2-$norm. For a function $f : \mathbb{R}^d \to \mathbb{R}$ we use $\nabla f$ to denote its gradient. For functions of two vector arguments, $f(\boldsymbol{x}, \boldsymbol{y}) : \mathbb{R}^{d_1} \times \mathbb{R}^{d_2} \to \mathbb{R}$, we use $\nabla_{\boldsymbol{x}} f, \nabla_{\boldsymbol{y}} f$ to denote its partial gradient. For the time derivative we will use the dot accent abbreviation, i.e., $\dot{\boldsymbol{x}} = \frac{d}{dt}[\boldsymbol{x}(t)]$. A function $f$ will belong to $C^r$ if it is $r$ times continuously differentiable. The term "sigmoid" function refers to $\sigma : \mathbb{R} \to \mathbb{R}$ such that $\sigma(x) = (1 + e^{-x})^{-1}$. Finally, we use $\mathcal{P}(\cdot)$, operating over a set, to denote its (Lebesgue) measure.

### 3.2 Definitions

**Definition 1** (Hidden Bilinear Zero-Sum Game). *In a hidden bilinear zero-sum game there are two players, each one equipped with a smooth function $\boldsymbol{F} : \mathbb{R}^n \to \mathbb{R}^N$ and $\boldsymbol{G} : \mathbb{R}^m \to \mathbb{R}^M$ and a payoff matrix $U_{N \times M}$ such that each player inputs its own decision vector $\boldsymbol{\theta} \in \mathbb{R}^n$ and $\boldsymbol{\phi} \in \mathbb{R}^m$ and is trying to maximize or minimize $r(\boldsymbol{\theta}, \boldsymbol{\phi}) = \boldsymbol{F}(\boldsymbol{\theta})^\top U \boldsymbol{G}(\boldsymbol{\phi})$ respectively.*

In this work we will mostly study continuous time dynamics of solutions for the problem of Equation 1 for hidden bilinear zero-sum games but we will also make some important connections to discrete time dynamics that are also prevalent in practice. In order to make this distinction clear, let us define the following terms.

**Definition 2** (Continuous Time Dynamical System). *A system of ordinary differential equations $\dot{\boldsymbol{x}} = f(\boldsymbol{x})$ where $f : \mathbb{R}^d \to \mathbb{R}^d$ will be called a continuous time dynamical system. Solutions of the equation $f(\boldsymbol{x}) = 0$ are called the fixed points of the dynamical system.*

We will call $f$ the *vector field* of the dynamical system. In order to understand the properties of continuous time dynamical systems, we will often need to study their behaviour given different initial conditions. This behaviour is captured by the flow of the dynamical system. More precisely,

**Definition 3.** *If $f$ is Lipschitz-continuous, there exists a continuous map $\Phi(\boldsymbol{x}_0, t) : \mathbb{R}^d \times \mathbb{R} \to \mathbb{R}^d$ called* flow *of the dynamical system such that for all $\boldsymbol{x}_0 \in \mathbb{R}^d$ we have that $\Phi(\boldsymbol{x}_0, t)$ is the unique solution of the problem $\{\dot{\boldsymbol{x}} = f(\boldsymbol{x}), \boldsymbol{x}(0) = \boldsymbol{x}_0\}$. We will refer to $\Phi(\boldsymbol{x}_0, t)$ as a* trajectory *or* orbit *of the dynamical system.*

In this work we will be mainly study the gradient-descent-ascent dynamics for the problem of Equation 1. The continuous (discrete) time version of the dynamics (with learning rate $\alpha$) are based on the following equations:

$$(\text{CGDA}) : \begin{Bmatrix} \dot{\boldsymbol{\theta}} & = -\nabla_{\boldsymbol{\theta}} r(\boldsymbol{\theta}, \boldsymbol{\phi}) \\ \dot{\boldsymbol{\phi}} & = \nabla_{\boldsymbol{\phi}} r(\boldsymbol{\theta}, \boldsymbol{\phi}) \end{Bmatrix} \quad (\text{DGDA}) : \begin{Bmatrix} \boldsymbol{\theta}_{k+1} & = \boldsymbol{\theta}_k - \alpha \nabla_{\boldsymbol{\theta}} r(\boldsymbol{\theta}_k, \boldsymbol{\phi}_k) \\ \boldsymbol{\phi}_{k+1} & = \boldsymbol{\phi}_k + \alpha \nabla_{\boldsymbol{\phi}} r(\boldsymbol{\theta}_k, \boldsymbol{\phi}_k) \end{Bmatrix}$$

A key notion in our analysis is that of (Poincaré) recurrence. Intuitively, a dynamical system is recurrent if, after a sufficiently long (but finite) time, almost every state returns arbitrarily close to the system's initial state.

**Definition 4.** *A point $\mathbf{x} \in \mathbb{R}^d$ is said to be recurrent under the flow $\Phi$, if for every neighborhood $U \subseteq \mathbb{R}^d$ of $\mathbf{x}$, there exists an increasing sequence of times $t_n$ such that $\lim_{n \to \infty} t_n = \infty$ and $\Phi(\mathbf{x}, t_n) \in U$ for all $n$. Moreover, the flow $\Phi$ is called Poincaré recurrent in non-zero measure set $A \subseteq \mathbb{R}^d$ if the set of the non-recurrent points in $A$ has zero measure.*

## 4 Cycles in hidden bilinear games with two strategies

In this section we will focus on a particular case of hidden biinear games where both the generator and the discriminator play only two strategies. Let $U$ be our zero-sum game and without loss of generality we can assume that there are functions $f : \mathbb{R}^n \to [0, 1]$ and $g : \mathbb{R}^m \to [0, 1]$ such that

$$\boldsymbol{F}(\boldsymbol{\theta}) = \begin{pmatrix} f(\boldsymbol{\theta}) \\ 1 - f(\boldsymbol{\theta}) \end{pmatrix} \qquad U = \begin{pmatrix} u_{0,0} & u_{0,1} \\ u_{1,0} & u_{1,1} \end{pmatrix} \qquad \boldsymbol{G}(\boldsymbol{\phi}) = \begin{pmatrix} g(\boldsymbol{\phi}) \\ 1 - g(\boldsymbol{\phi}) \end{pmatrix}$$

Let us assume that the hidden bi-linear game has a unique mixed Nash equilibrium $(p, q)$:

$$v = u_{0,0} - u_{0,1} - u_{1,0} + u_{1,1} \neq 0, \quad q = -\frac{u_{0,1} - u_{1,1}}{v} \in (0, 1), \quad p = -\frac{u_{1,0} - u_{1,1}}{v} \in (0, 1)$$

Then we can write down the equations of gradient-descent-ascent : $\begin{Bmatrix} \dot{\boldsymbol{\theta}} = -v\nabla f(\boldsymbol{\theta})(g(\boldsymbol{\phi}) - q) \\ \dot{\boldsymbol{\phi}} = v\nabla g(\boldsymbol{\phi})(f(\boldsymbol{\theta}) - p) \end{Bmatrix}$ (3)

In order to analyze the behavior of this system, we would like to understand the topology of the trajectories of $\boldsymbol{\theta}$ and $\boldsymbol{\phi}$, at least individually. The following lemma makes a connection between the trajectories of each variable in the min-max optimization system of Equation 3 and simple gradient ascent dynamics.

**Lemma 1.** *Let $k : \mathbb{R}^d \to \mathbb{R}$ be a $C^2$ function. Let $h : \mathbb{R} \to \mathbb{R}$ be a $C^1$ function and $\boldsymbol{x}(t) = \rho(t)$ be the unique solution of the dynamical system $\Sigma_1$. Then for the dynamical system $\Sigma_2$ the unique solution is $\boldsymbol{z}(t) = \rho(\int_0^t h(s)\mathrm{d}s)$*

$$\begin{Bmatrix} \dot{\boldsymbol{x}} & = & \nabla k(\boldsymbol{x}) \\ \boldsymbol{x}(0) & = & \boldsymbol{x}_0 \end{Bmatrix} : \Sigma_1 \quad \begin{Bmatrix} \dot{\boldsymbol{z}} & = & h(t)\nabla k(\boldsymbol{z}) \\ \boldsymbol{z}(0) & = & \boldsymbol{x}_0 \end{Bmatrix} : \Sigma_2$$

By applying the previous result for $\boldsymbol{\theta}$ with $k = f$ and $h(t) = -v(g(\boldsymbol{\phi}(t)) - q)$, we get that even under the dynamics of Equation 3, $\boldsymbol{\theta}$ remains on a trajectory of the simple gradient ascent dynamics with initial condition $\boldsymbol{\theta}(0)$. This necessarily affects the possible values of $f$ and $g$ given the initial conditions. Let us define the sets of values attainable for each initialization.

**Definition 5.** *For each $\boldsymbol{\theta}(0)$, $f_{\boldsymbol{\theta}(0)}$ is the set of possible values of $f(\boldsymbol{\theta}(t))$ can attain under gradient ascent dynamics. Similarly, we define $g_{\boldsymbol{\phi}(0)}$ the corresponding set for g.*

What is special about the trajectories of gradient ascent is that along this curve $f$ is strictly increasing (For a detailed explanation, reader could check the proof of Theorem 1 in the Appendix) and therefore each point $\boldsymbol{\theta}(t)$ in the trajectory has a unique value for $f$. Therefore even in the system of Equation 3, $f(\boldsymbol{\theta}(t))$ uniquely identifies $\boldsymbol{\theta}(t)$. This can be formalized in the next theorem.

**Theorem 1.** *For each $\boldsymbol{\theta}(0), \boldsymbol{\phi}(0)$, under the dynamics of Equation 3, there are $C^1$ functions $(X_{\boldsymbol{\theta}(0)}, X_{\boldsymbol{\phi}(0)})$ such that $X_{\boldsymbol{\theta}(0)} : f_{\boldsymbol{\theta}(0)} \rightarrow \mathbb{R}^n, X_{\boldsymbol{\phi}(0)} : g_{\boldsymbol{\phi}(0)} \rightarrow \mathbb{R}^n$ and $\boldsymbol{\theta}(t) = X_{\boldsymbol{\theta}(0)}(f(t))$, $\boldsymbol{\phi}(t) = X_{\boldsymbol{\phi}(0)}(g(t))$.*

Equipped with these results, we are able to reduce this complicated dynamical system of $\boldsymbol{\theta}$ and $\boldsymbol{\phi}$ to a planar dynamical system involving $f$ and $g$ alone.

**Lemma 2.** *If $\boldsymbol{\theta}(t)$ and $\boldsymbol{\phi}(t)$ are solutions to Equation 3 with initial conditions $(\boldsymbol{\theta}(0), \boldsymbol{\phi}(0))$, then we have that $f(t) = f(\boldsymbol{\theta}(t))$ and $g(t) = g(\boldsymbol{\phi}(t))$ satisfy the following equations*

$$\begin{aligned} \dot{f} &= -v\|\nabla f(X_{\boldsymbol{\theta}(0)}(f))\|^2(g-q) \\ \dot{g} &= v\|\nabla g(X_{\boldsymbol{\phi}(0)}(g))\|^2(f-p) \end{aligned} \tag{4}$$

As one can observe both form Equation 3 and Equation 4, fixed points of the gradient-descent-ascent dynamics correspond to either solutions of $f(\boldsymbol{\theta}) = p$ and $g(\boldsymbol{\phi}) = q$ or stationary points of $f$ and $g$ or even some combinations of the aforementioned conditions. Although, all of them are fixed points of the dynamical system, only the former equilibria are game theoretically meaningful. We will therefore define a subset of initial conditions for Equation 3 such that convergence to game theoretically meaningful fixed points may actually be feasible:

**Definition 6.** *We will call the initialization $(\boldsymbol{\theta}(0), \boldsymbol{\phi}(0))$ safe for Equation 3 if $\boldsymbol{\theta}(0)$ and $\boldsymbol{\phi}(0)$ are not stationary points of $f$ and $g$ respectively and $p \in f_{\boldsymbol{\theta}(0)}$ and $q \in g_{\boldsymbol{\phi}(0)}$.*

For safe initial conditions we can show that gradient-descent-ascent dynamics applied in the class of the hidden bilinear zero-sum game mimic properties and behaviors of conservative/Hamiltonian physical systems [BP19b], like an ideal pendulum or an ideal spring-mass system. In such systems, there is a notion of energy that remains constant over time and hence the system trajectories lie on level sets of these functions. To motivate further this intuition, it is easy to check that for the simplified case where $\|\nabla f\| = \|\nabla g\| = 1$ the level sets correspond to cycles centered at the Nash equilibrium and the system as a whole captures gradient-descent-ascent for a bilinear $2 \times 2$ zero-sum game (e.g. Matching Pennies).

**Theorem 2.** *Let $\boldsymbol{\theta}(0)$ and $\boldsymbol{\phi}(0)$ be safe initial conditions. Then for the system of Equation 3, the following quantity is time-invariant*

$$H(f,g) = \int_p^f \frac{z-p}{\|\nabla f(X_{\boldsymbol{\theta}(0)}(z))\|^2}\mathrm{d}z + \int_q^g \frac{z-q}{\|\nabla g(X_{\boldsymbol{\phi}(0)}(z))\|^2}\mathrm{d}z$$

The existence of this invariant immediately guarantees that Nash Equilibrium $(p,q)$ cannot be reached if the dynamical system is not initialized there. Taking advantage of the planarity of the induced system - a necessary condition of Poincaré-Bendixson Theorem - we can prove that:

**Theorem 3.** *Let $\boldsymbol{\theta}(0)$ and $\boldsymbol{\phi}(0)$ be safe initial conditions. Then for the system of Equation 3, the orbit $(\boldsymbol{\theta}(t), \boldsymbol{\phi}(t))$ is periodic.*

On a positive note, we can prove that the time averages of $f$ and $g$ as well as the time averages of expected utilities of both players converge to their Nash equilibrium values.

**Theorem 4.** *Let $\boldsymbol{\theta}(0)$ and $\boldsymbol{\phi}(0)$ be safe initial conditions and $(\boldsymbol{P}, \boldsymbol{Q}) = \left(\binom{p}{1-p}, \binom{q}{1-q}\right)$, then for the system of Equation 3*

$$\lim_{T \to \infty} \frac{\int_0^T f(\boldsymbol{\theta}(t))\mathrm{d}t}{T} = p, \quad \lim_{T \to \infty} \frac{\int_0^T r(\boldsymbol{\theta}(t), \boldsymbol{\phi}(t))\mathrm{d}t}{T} = \boldsymbol{P}^\top U \boldsymbol{Q}, \quad \lim_{T \to \infty} \frac{\int_0^T g(\boldsymbol{\phi}(t))\mathrm{d}t}{T} = q$$

# 5 Poincaré recurrence in hidden bilinear games with more strategies

In this section we will extend our results by allowing both the generator and the discriminator to play hidden bilinear games with more than two strategies. We will specifically study the case of hidden bilinear games where each coordinate of the vector valued functions $F$ and $G$ is controlled by disjoint subsets of the variables $\boldsymbol{\theta}$ and $\boldsymbol{\phi}$, i.e.

$$\boldsymbol{\theta} = \begin{bmatrix} \boldsymbol{\theta}_1 \\ \boldsymbol{\theta}_2 \\ \vdots \\ \boldsymbol{\theta}_N \end{bmatrix} \quad \boldsymbol{F}(\boldsymbol{\theta}) = \begin{bmatrix} f_1(\boldsymbol{\theta}_1) \\ f_2(\boldsymbol{\theta}_2) \\ \vdots \\ f_N(\boldsymbol{\theta}_N) \end{bmatrix} \quad \boldsymbol{\phi} = \begin{bmatrix} \boldsymbol{\phi}_1 \\ \boldsymbol{\phi}_2 \\ \vdots \\ \boldsymbol{\phi}_M \end{bmatrix} \quad \boldsymbol{G}(\boldsymbol{\phi}) = \begin{bmatrix} g_1(\boldsymbol{\phi}_1) \\ g_2(\boldsymbol{\phi}_2) \\ \vdots \\ g_M(\boldsymbol{\phi}_M) \end{bmatrix} \tag{5}$$

where each function $f_i$ and $g_i$ takes an appropriately sized vector and returns a non-negative number. To account for possible constraints (e.g. that probabilities of each distribution must sum to one), we will incorporate this restriction using Lagrange Multipliers. The resulting problem becomes

$$\min_{\boldsymbol{\theta} \in \mathbb{R}^n, \mu \in \mathbb{R}} \max_{\boldsymbol{\phi} \in \mathbb{R}^m, \lambda \in \mathbb{R}} \boldsymbol{F}(\boldsymbol{\theta})^\top U \boldsymbol{G}(\boldsymbol{\phi}) + \lambda \left( \sum_{i=1}^N f_i(\boldsymbol{\theta}_i) - 1 \right) + \mu \left( \sum_{i=j}^M g_j(\boldsymbol{\phi}_j) - 1 \right) \tag{6}$$

Writing down the equations of gradient-ascent-descent we get

$$
\begin{aligned}
\dot{\boldsymbol{\theta}}_i &= -\nabla f_i(\boldsymbol{\theta}_i) \left( \sum_{j=1}^M u_{i,j} g_j(\boldsymbol{\phi}_j) + \lambda \right) \quad & \dot{\boldsymbol{\phi}}_j &= \nabla g_j(\boldsymbol{\phi}_j) \left( \sum_{i=1}^N u_{i,j} f_i(\boldsymbol{\theta}_i) + \mu \right) \\
\dot{\mu} &= -\left( \sum_{j=1}^M g_j(\boldsymbol{\phi}_j) - 1 \right) & \dot{\lambda} &= \left( \sum_{i=1}^N f_i(\boldsymbol{\theta}_i) - 1 \right)
\end{aligned}
\tag{7}
$$

Once again we can show that along the trajectories of the system of Equation 7, $\boldsymbol{\theta}_i$ can be uniquely identified by $f_i(\boldsymbol{\theta}_i)$ given $\boldsymbol{\theta}_i(0)$ and the same holds for the discriminator. This allows us to construct functions $X_{\boldsymbol{\theta}_i(0)}$ and $X_{\boldsymbol{\phi}_j(0)}$ just like in Theorem 1. We can now write down a dynamical system involving only $f_i$ and $g_j$.

**Lemma 3.** *If $\boldsymbol{\theta}(t)$ and $\boldsymbol{\phi}(t)$ are solutions to Equation 7 with initial conditions $(\boldsymbol{\theta}(0), \boldsymbol{\phi}(0), \lambda(0), \mu(0))$, then we have that $f_i(t) = f_i(\boldsymbol{\theta}_i(t))$ and $g_j(t) = g_j(\boldsymbol{\phi}_j(t))$ satisfy the following equations*

$$
\begin{aligned}
\dot{f}_i &= -\|\nabla f_i(X_{\boldsymbol{\theta}_i(0)}(f_i))\|^2 \left( \sum_{j=1}^M u_{i,j} g_j + \lambda \right) \\
\dot{g}_j &= \|\nabla g_j(X_{\boldsymbol{\phi}_j(0)}(g_j))\|^2 \left( \sum_{i=1}^N u_{i,j} f_i + \mu \right)
\end{aligned}
\tag{8}
$$

Similarly to the previous section, we can define a notion of safety for Equation 7. Let us assume that the hidden Game has a fully mixed Nash equilibrium $(\boldsymbol{p}, \boldsymbol{q})$. Then we can define

**Definition 7.** *We will call the initialization $(\boldsymbol{\theta}(0), \boldsymbol{\phi}(0), \lambda(0), \mu(0))$ safe for Equation 7 if $\boldsymbol{\theta}_i(0)$ and $\boldsymbol{\phi}_j(0)$ are not stationary points of $f_i$ and $g_j$ respectively and $p_i \in f_{i_{\boldsymbol{\theta}_i(0)}}$ and $q_j \in g_{j_{\boldsymbol{\phi}_j(0)}}$.*

**Theorem 5.** *Assume that $(\boldsymbol{\theta}(0), \boldsymbol{\phi}(0), \lambda(0), \mu(0))$ is a safe initialization. Then there exist $\lambda_*$ and $\mu_*$ such that the following quantity is time invariant:*

$$H(\boldsymbol{F}, \boldsymbol{G}, \lambda, \mu) = \sum_{i=1}^N \int_{p_i}^{f_i} \frac{z - p_i}{\|\nabla f_i(X_{\boldsymbol{\theta}_i(0)}(z))\|^2} \mathrm{d}z + \sum_{j=1}^M \int_{q_j}^{g_j} \frac{z - q_j}{\|\nabla g_j(X_{\boldsymbol{\phi}_j(0)}(z))\|^2} \mathrm{d}z +$$

$$\int_{\lambda^*}^\lambda (z - \lambda^*) \, \mathrm{d}z + \int_{\mu^*}^\mu (z - \mu^*) \, \mathrm{d}z$$

Given that even our reduced dynamical system has more than two state variables we cannot apply the Poincaré-Bendixson Theorem. Instead we can prove that there exists a one to one differentiable

transformation of our dynamical system so that the resulting system becomes divergence free. Applying Louville's formula, the flow of the the transformed system is volume preserving. Combined with the invariant of Theorem 5, we can prove that the variables of the transformed system remain bounded. This gives us the following guarantees

**Theorem 6.** *Assume that $(\boldsymbol{\theta}(0), \boldsymbol{\phi}(0), \lambda(0), \mu(0))$ is a safe initialization. Then the trajectory under the dynamics of Equation 7 is diffeomoprphic to one trajectory of a Poincaré recurrent flow.*

This result implies that if the corresponding trajectory of the Poincaré recurrent flow is itself recurrent, which almost all of them are, then the trajectory of the dynamics of Equation 7 is also recurrent. This is however not enough to reason about how often any of the trajectories of the dynamics of Equation 7 is recurrent. In order to prove that the flow of Equation 7 is Poincaré recurrent we will make some additional assumptions

**Theorem 7.** *Let $f_i$ and $g_j$ be sigmoid functions. Then the flow of Equation 7 is Poincaré recurrent. The same holds for all functions $f_i$ and $g_j$ that are one to one functions and for which all initializations are safe.*

It is worth noting that for the unconstrained version of the previous min-max problem we arrive at the same conclusions/theorems by repeating the above analysis without using the Lagrange multipliers.

## 6    Spurious equilibria

In the previous sections we have analyzed the behavior of safe initializations and we have proved that they lead to either periodic or recurrent trajectories. For initializations that are not safe for some equilibrium of the hidden game, game theoretically interesting fixed points are not even realizable solutions. In fact we can prove something stronger:

**Theorem 8.** *One can construct functions $f$ and $g$ for the system of Equation 3 so that for a positive measure set of initial conditions the trajectories converge to fixed points that do not correspond to equilibria of the hidden game.*

The main idea behind our theorem is that we can construct functions $f$ and $g$ that have local optima that break the safety assumption. For a careful choice of the value of the local optima we can make these fixed points stable and then the Stable Manifold Theorem guarantees that a non zero measure set of points in the vicinity of the fixed point converges to it. Of course the idea of these constructions can be extended to our analysis of hidden games with more strategies.

## 7    Discrete Time Gradient-Ascent-Descent

In this section we will discuss the implications of our analysis of continuous time gradient-ascent-descent dynamics on the properties of their discrete time counterparts. In general, the behavior of discrete time dynamical systems can be significantly different [LY75, BP18, PPP17] so it is critical to perform this non-trivial analysis. We are able to show that the picture of non-equilibriation persists for an interesting class of hidden bilinear games.

**Theorem 9.** *Let $f_i$ and $g_j$ be sigmoid functions. Then for the discretized version of the system of Equation 7 and for safe intializations, function $H$ of Theorem 5 is non-decreasing.*

An immediate consequence of the above theorem is that the discretized system cannot converge to the equilibrium $(\boldsymbol{p}, \boldsymbol{q})$ if its not initialized there. For the case of non-safe initializations, the conclusions of Theorem 8 persist in this case as well.

**Theorem 10.** *One can choose a learning rate $\alpha$ and functions $f$ and $g$ for the discretized version of the system of Equation 3 so that for a positive measure set of initial conditions the trajectories converge to fixed points that do not correspond to equilibria of the hidden game.*

## 8    Conclusion

In this work, inspired broadly by the structure of the complex competition between generators and discriminators in GANs, we defined a broad class of non-convex non-concave min max optimization

games, which we call hidden bilinear zero-sum games. In this setting, we showed that gradient-descent-ascent behavior is considerably more complex than a straightforward convergence to the min-max solution that one might at first suspect. We showed that the trajectories even for the simplest but evocative 2x2 game exhibits cycles. In higher dimensional games, the induced dynamical system could exhibit even more complex behavior like Poincare recurrence. On the other hand, we explored safety conditions whose violation may result in convergence to spurious game-theoretically meaningless equilibria. Finally, we show that even for a simple but widespread family of functions like sigmoids discretizing gradient-descent-ascent can further intensify the disequilibrium phenomena resulting in divergence away from equilibrium.

As a consequence of this work numerous open problems emerge; Firstly, extending such recurrence results to more general families of functions, as well as examining possible generalizations to multi-player network zero-sum games are fascinating questions. Recently, there has been some progress in resolving cyclic behavior in simpler settings by employing different training algorithms/dynamics (e.g., [DISZ18, MLZ+19b, GHP+19b]). It would be interesting to examine if these algorithms could enhance equilibration in our setting as well. Additionally, the proposed safety conditions shows that a major source of spurious equilibria in GANs could be the bad local optima of the individual neural networks of the discriminator and the generator. Lessons learned from overparametrized neural network architectures that converge to global optima [DLL+18] could lead to improved efficiency in training GANs. Finally, analyzing different simplification/models of GANs where provable convergence is possible could lead to interesting comparisons as well as to the emergence of theoretically tractable hybrid models that capture both the hardness of GAN training (e.g. non-convergence, cycling, spurious equilibria, mode collapse, etc) as well as their power.

## Acknowledgements

Georgios Piliouras acknowledges MOE AcRF Tier 2 Grant 2016-T2-1-170, grant PIE-SGP-AI-2018-01 and NRF 2018 Fellowship NRF-NRFF2018-07. Emmanouil-Vasileios Vlatakis-Gkaragkounis was supported by NSF CCF-1563155, NSF CCF-1814873, NSF CCF-1703925, NSF CCF-1763970. Finally this work was supported by the Onassis Foundation - Scholarship ID: F ZN 010-1/2017-2018.

## Footnotes

[2]Interestingly, running alternating gradient-descent-ascent in discrete-time results once again in recurrent behavior [BGP19].

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
