[Supplementary Material]

# Poincaré Recurrence, Cycles and Spurious Equilibria in Gradient-Descent-Ascent for Non-Convex Non-Concave Zero-Sum Games

## Supplementary Material

## 1 Background in dynamical systems

### 1.1 Poincaré-Bendixson Theorem

The Poincaré-Bendixson theorem is a powerful theorem that implies that two-dimensional systems cannot exhibit chaos. Effectively, the limit behavior is either going to be an equilibrium, a periodic orbit, or a closed loop, punctuated by one (or more) fixed points. Formally, we have:

**Theorem 1** (Poincaré-Bendixson Theorem [Ben01]). *Given a differentiable real dynamical system defined on an open subset of the plane, then every non-empty compact $\omega$-limit set of an orbit, which contains only finitely many fixed points, is either a fixed point, a periodic orbit, or a connected set composed of a finite number of fixed points together with homoclinic and heteroclinic orbits connecting these.*

### 1.2 Liouville's formula and Poincaré recurrence

In order to study the flows of dynamical systems in higher dimensions, one needs to understand more about the behaviour of the flow $\Phi$ both in time and space. An important property is the evolution of the volume of $\Phi$ over time:

**Theorem 2** (Liouville's formula). *Let $\Phi$ be the flow of a dynamical system with vecor field $f$. Given any measurable set $A$, let $A(t) = \Phi(A, t)$ and its volume be $\mathrm{vol}[A(t)] = \int_{A(t)} \mathrm{d}\boldsymbol{x}$. Then we have that*

$$\frac{d\mathrm{vol}[A(t)]}{\mathrm{d}t} = \int_{A(t)} \mathrm{div}[f(\boldsymbol{x})]\mathrm{d}\boldsymbol{x}$$

An interesting class of dynamical systems are those whose vector fields have zero divergence everywhere. Liouville's formula trivially implies that the volume of the flow is preserved in such systems. This is an important tool for proving that a flow of a dynamical system is Poincaré recurrent.

**Theorem 3** (Poincaré Recurrence Theorem (version 1) [Poi90a]). *Let $(X, \Sigma, \mu)$ be a finite measure space and let $f \colon X \to X$ be a measure-preserving transformation. Then, for any $E \in \Sigma$, the set of those points $x$ of $E$ such that $f^n(x) \notin E$ for all $n > 0$ has zero measure. That is, almost every point of $E$ returns to $E$. In fact, almost every point returns infinitely often. Namely,*

$$\mathcal{P}\left(\{x \in E : \exists N \text{ such that } f^n(x) \notin E \text{ for all } n > N\}\right) = 0.$$

[Poi90a] proved that in certain systems almost all trajectories return arbitrarily close to their initial position infinitely often. Indeed, let $f \colon X \to X$ be a measure-preserving transformation, $\{U_n : n \in \mathbb{N}\}$ be a basis of open sets for the bounded subset $X \subset \mathbb{R}^d$, and for each $n$ define $U_{n'} = \{x \in U_n : \forall n \geq 1, f_n(x) \notin U_n\}$. Notice that such basis exists since $\mathbb{R}^n$ is a second-countable Hausdorff space. From the initial theorem we know that $\mathcal{P}(U_{n'}) = 0$. Let $\mathcal{U} = \cup_{n \in \mathbb{N}} U_{n'}$. Then $\mathcal{P}(\mathcal{U}) = 0$. We assert that if $x \in X \setminus \mathcal{U}$ then $x$ is recurrent. In fact, given a neighborhood $U$ of $x$, there is a basic neighborhood $U_n$ such that $\{x\} \subset U_n \subset U$, and since $x \notin \mathcal{U}$ we have that $x \in U_n \setminus U_{n'}$ which by definition of $U_{n'}$ means that there exists $n \geq 1$ such that $f_n(x) \in U_n \subset U$. Thus $x$ is recurrent. Therefore, for the rest of the paper, we will use the following version which is common in dynamical systems nomenclature.

**Theorem 4** (Poincaré Recurrence Theorem (dynamical system version)). *[Poi90b] If a flow $\Phi :$ $\mathbb{R}^n \times \mathbb{R} \to \mathbb{R}^n$ preserves volume and has only orbits on a bounded subset $D$ of $\mathbb{R}^n$ then almost each point in $D$ is recurrent, i.e for every open neighborhood $U$ of $x$ there exists an increasing sequence of times $t_n$ such that $\lim\limits_{n \to \infty} t_n = \infty$ and $\Phi(\mathbf{x}, t_n) \in U$ for all $n$.*

## 1.3 Additional Definitions

**Definition 1** (Differomorphism, [Per91]). *Let $U, V$ be manifolds. A map $f : U \to V$ is called a diffeomorphism if $f$ carries $U$ onto $V$ and also both $f$ and $f^{-1}$ are smooth.*

**Definition 2** (Topological conjugacy, [Per91]). *Two flows $\Phi_t : A \to A$ and $\Psi_t : B \to B$ are conjugate if there exists a homeomorphism $g : A \to B$ such that*

$$\forall \boldsymbol{x} \in A, t \in \mathbb{R} : g(\Phi_t(\boldsymbol{x})) = \Psi_t(g(\boldsymbol{x}))$$

*Furthermore, two flows $\Phi_t : A \to A$ and $\Psi_t : B \to B$ are diffeomorphic if there exists a diffeomorphism $g : A \to B$ such that*

$$\forall \boldsymbol{x} \in A, t \in \mathbb{R} : g(\Phi_t(\boldsymbol{x})) = \Psi_t(g(\boldsymbol{x})).$$

*If two flows are diffeomorphic, then their vector fields are related by the derivative of the conjugacy. That is, we get precisely the same result that we would have obtained if we simply transformed the coordinates in their differential equations*

**Definition 3** (($\alpha, \omega$)-limit set, [Per91]). *Let $\Phi(\boldsymbol{x}_0, \cdot)$ be the flow of an autonomous dynamical system $\dot{pmb{x}} = f(\boldsymbol{x})$. Then*

$$
\begin{aligned}
\omega(\boldsymbol{x_0}) = \quad & \{\boldsymbol{x} : \text{for all } T \text{ and all } \epsilon > 0 \text{ there exists } t > T \text{ such that } |\Phi(\boldsymbol{x_0}, t) - \boldsymbol{x}| < \epsilon\} \\
\alpha(\boldsymbol{x_0}) = \quad & \{\boldsymbol{x} : \text{for all } T \text{ and all } \epsilon > 0 \text{ there exists } t < T \text{ such that } |\Phi(\boldsymbol{x_0}, t) - \boldsymbol{x}| < \epsilon\}
\end{aligned}
$$

*Equivalently,*

$$
\omega(\boldsymbol{x_0}) = \quad \{\boldsymbol{x} : \text{there exists an unbounded, increasing sequence } \{t_k\} \text{ such that } \lim\limits_{k \to \infty} \Phi(t_k, \boldsymbol{x_0}) = \boldsymbol{x}\}
$$
$$
\alpha(\boldsymbol{x_0}) = \quad \{\boldsymbol{x} : \text{there exists an unbounded, decreasing sequence } \{t_k\} \text{ such that } \lim\limits_{k \to \infty} \Phi(t_k, \boldsymbol{x_0}) = \boldsymbol{x}\}
$$

**Lemma 1** (Recurrence and Conjugacy [MPP18]). *Let $\Phi_t : A \to A$ and $\Psi_t : B \to B$ be conjugate flows and $\gamma$ be the diffeomorphism which connects them. Then a point $\boldsymbol{x} \in V$ is recurrent for $\Phi$ if and only if $\gamma(\boldsymbol{x}) \in \gamma(V)$ is recurrent for $\Psi$.*

*Proof.* We will first prove the if direction. Let's take any open neighborhood $U \subseteq V$ around $\boldsymbol{x}$. Using the diffeomorphism, there is a unique $\gamma(U) \subseteq \gamma(V)$ and additionally since $U$ is open $\gamma(U)$ is also open. Obviously, $\gamma(\boldsymbol{x}) \in \gamma(U)$. Thus, if $\gamma(\boldsymbol{x})$ is recurrent there is an unbounded increasing sequence of moments $t_n$ such that

$$\Psi(\gamma(\boldsymbol{x}), t_n) \in \gamma(U).$$

This is equivalent with the fact that there is an unbounded increasing sequence of moments $t_n$ such that

$$\gamma^{-1}(\Psi(\gamma(\boldsymbol{x}), t_n)) \in \gamma^{-1}(\gamma(U)).$$

Using the basic property of topological conjugacy, we have that

$$\Phi(\boldsymbol{x}, t_n) = \gamma^{-1}(\Psi(\gamma(U), t_n)).$$

Thus, for $t_n$ we have that

$$\Phi(\boldsymbol{x}, t_n) \in U.$$

It follows that $\boldsymbol{x}$ is also recurrent for $\Phi$. The result for the opposite direction follows immediately by using the inverse map. $\square$

## 1.4 Stable Manifold Theorems

**Theorem 5** (Stable Manifold Theorem for Continuous Time Dynamical Systems p.120 [Per91])**.** *Let $E$ be an open subset of $\mathbb{R}^n$ containing the origin, let $f \in C^1(E)$, and let $\phi_t$ be the flow of the nonlinear system $\dot{\boldsymbol{x}} = f(\boldsymbol{x})$. Suppose that $f(\boldsymbol{0}) = \boldsymbol{0}$ and that $Df(\mathbf{O})$ has $k$ eigenvalues with negative real part and $n - k$ eigenvalues with positive real part. Then there exists a $k$-dimensional differentiable manifold $S$ tangent to the stable subspace $E^s$ of the linear system $\dot{\boldsymbol{x}} = Df(\boldsymbol{0})\boldsymbol{x}$ at $\boldsymbol{0}$ such that for all $t \geq 0$, $\phi_t(S) \subseteq S$ and for all $\boldsymbol{x}_0 \in S$:*

$$\lim_{t \to \infty} \phi_t(\boldsymbol{x}_0) = \boldsymbol{0}$$

*and there exists an $n - k$ dimensional differentiable manifold $U$ tangent to the unstable subspace $E^u$ of the linear system $\dot{\boldsymbol{x}} = Df(\boldsymbol{0})\boldsymbol{x}$ at $\boldsymbol{0}$ such that for all $t \leq 0$, $\phi_t(U) \subseteq U$ and for all $\boldsymbol{x}_0 \in U$:*

$$\lim_{t \to -\infty} \phi_t(\boldsymbol{x}_0) = \boldsymbol{0}$$

**Theorem 6** (Center and Stable Manifolds, p. 65 of [Shu87])**.** *Let $\boldsymbol{p}$ be a fixed point for the $C^r$ local diffeomorphism $h : U \to \mathbb{R}^n$ where $U \subset \mathbb{R}^n$ is an open neighborhood of $\boldsymbol{p}$ in $\mathbb{R}^n$ and $r \geq 1$. Let $E^s \oplus E^c \oplus E^u$ be the invariant splitting of $\mathbb{R}^n$ into generalized eigenspaces of $Dh(\boldsymbol{p})$[1] corresponding to eigenvalues of absolute value less than one, equal to one, and greater than one. To the $Dh(\boldsymbol{p})$ invariant subspace $E^s \oplus E^c$ there is an associated local $h$ invariant $C^r$ embedded disc $W_{loc}^{sc}$ of dimension $dim(E^s \oplus E^c)$, and ball $B$ around $\boldsymbol{p}$ such that:*

$$h(W_{loc}^{sc}) \cap B \subset W_{loc}^{sc}. \text{ If } h^n(\boldsymbol{x}) \in B \text{ for all } n \geq 0, \text{ then } \boldsymbol{x} \in W_{loc}^{sc}.$$

## 1.5 Regular Value Theorem

**Definition 4.** *Let $f : U \to V$ be a smooth map between same dimensional manifolds. We denote that $x \in U$ is a regular point if the derivative is nonsingular. $y \in V$ is called a **regular value** if $f^{-1}(y)$ contains only regular points. If the derivative is singular, then $x$ is called a **critical point**. We also say $y \in V$ is a critical value if $y$ is not a regular value.*

**Theorem 7** (Regular Value Theorem)**.** *If $y \in Y$ is a regular value of $f : X \to Y$ then $f^{-1}(y)$ is a manifold of dimension $n - m$, since $dim(X) = n$ and $dim(Y) = m$.*

## 2  Omitted Proofs of Section 4
## Warm up: Cycles in hidden bilinear games with two strategies

In this first section, we show a key technical lemma which will be used in many different parts of our proof. More specifically, it shows how someone can derive the solution for a non-autonomous system via a conjugate autonomous dynamical system. The main intuition is that if the non-autonomous term is multiplicative and common across all terms of a vector field then it dictates the magnitude of the vector field (the speed of the motion), but does not affect directionality other than moving backwards or forwards along the same trajectory.

**Lemma 2** (Restated Lemma 1). *Let $k : \mathbb{R}^d \to \mathbb{R}$ be a $C^2$ function. Let $h : \mathbb{R} \to \mathbb{R}$ be a $C^1$ function and $\boldsymbol{x}(t) = \rho(t)$ be the unique solution of the dynamical system $\Sigma_1$. Then for the dynamical system $\Sigma_2$ the unique solution is $\boldsymbol{z}(t) = \rho(\int_0^t h(s)\mathrm{d}s)$*

$$\left\{ \begin{matrix} \dot{\boldsymbol{x}} & = & \nabla k(\boldsymbol{x}) \\ \boldsymbol{x}(0) & = & \boldsymbol{x}_0 \end{matrix} \right\} : \Sigma_1 \quad \left\{ \begin{matrix} \dot{\boldsymbol{z}} & = & h(t)\nabla k(\boldsymbol{z}) \\ \boldsymbol{z}(0) & = & \boldsymbol{x}_0 \end{matrix} \right\} : \Sigma_2$$

*Proof.* Firstly, notice that it holds $\rho(0) = \boldsymbol{x}_0$ and $\dot{\rho} = \nabla k(\rho)$, since $\rho$ is the unique solution of $\Sigma_1$ It is easy to check that:

$$\boldsymbol{z}(0) = \rho(\int_0^0 h(s)\mathrm{d}s) = \rho(0) = \boldsymbol{x}_0$$

$$\dot{\boldsymbol{z}} = \nabla\rho(\int_0^t h(s)\mathrm{d}s) \times \frac{\mathrm{d}[\int_0^t h(s)\mathrm{d}s]}{\mathrm{d}t}$$

$$= \nabla\rho(\int_0^t h(s)\mathrm{d}s)h(t)$$

□

The next proposition states that initial condition $(\boldsymbol{\theta}(0), \boldsymbol{\phi}(0))$ as well as $\{f(t), g(t)\}_{t=0}^\infty$ are sufficient to derive the complete system state of Continuous GDA $(\theta_{\theta_0}(t), \phi_{\phi_0}(t))$. The importance of the below theorem arises when someone takes into consideration periodicity and recurrence phenomena. Due to the existence of mapping $(f(t), g(t))$ to a unique $(\boldsymbol{\theta}(t), \boldsymbol{\phi}(t))$ given some initial condition $(\boldsymbol{\theta}(0), \boldsymbol{\phi}(0))$, any periodic or recurrent behavior of $(f(t), g(t))$ extends to the system trajectories.

**Theorem 8** (Restated Theorem 1). *For each $\boldsymbol{\theta}(0), \boldsymbol{\phi}(0)$, under the dynamics of Equation 3, there are $C^1$ functions $(X_{\boldsymbol{\theta}(0)}, X_{\boldsymbol{\phi}(0)})$ such that $X_{\boldsymbol{\theta}(0)} : f_{\boldsymbol{\theta}(0)} \to \mathbb{R}^n$, $X_{\boldsymbol{\phi}(0)} : g_{\boldsymbol{\phi}(0)} \to \mathbb{R}^n$ and $\boldsymbol{\theta}(t) = X_{\boldsymbol{\theta}(0)}(f(t)), \boldsymbol{\phi}(t) = X_{\boldsymbol{\phi}(0)}(g(t))$.*

*Proof.* Let us first study a simpler dynamical system $(\Sigma^*)$ with unique solution of $\gamma_{\boldsymbol{\theta}(0)}(t)$.

$$(\Sigma^*) \equiv \left\{ \begin{matrix} \dot{\boldsymbol{\theta}} & = & \nabla f(\boldsymbol{\theta}) \\ \boldsymbol{\theta}(0) & = & \boldsymbol{\theta}_0 \end{matrix} \right\}$$

It is easy to observe that:

$$\dot{f} = \nabla f(\boldsymbol{\theta})\dot{\boldsymbol{\theta}} = \|\nabla f(\boldsymbol{\theta})\|^2$$

If $\boldsymbol{x}_0$ is a stationary point of $f$ then the trajectory is a single point and the theorem holds trivially. If $\boldsymbol{x}_0$ is not a stationary point of $f$, $f$ continuously increases along the trajectory of the dynamical system. Therefore $A_{\boldsymbol{\theta}(0)}(t) = f(\gamma_{\boldsymbol{x}_0}(t))$ is an increasing function and therefore invertible. Let us call $A_{\boldsymbol{\theta}(0)}^{-1}(f)$ the inverse.

Let's recall now the dynamical system of our interest ( Equation 3 )

$$\text{CGDA} : \left\{ \begin{matrix} \dot{\boldsymbol{\theta}} = -v\nabla f(\boldsymbol{\theta})(g(\boldsymbol{\phi}) - q) \\ \dot{\boldsymbol{\phi}} = v\nabla g(\boldsymbol{\phi})(f(\boldsymbol{\theta}) - p) \end{matrix} \right\}$$

and more precisely to the $\boldsymbol{\theta}$-part of the system,i.e

$$(\Sigma) \equiv \begin{cases} \dot{\boldsymbol{\theta}} & = & -v\nabla f(\boldsymbol{\theta})(g(\boldsymbol{\phi}) - q) \\ \boldsymbol{\theta}(0) & = & \boldsymbol{\theta}_0 \end{cases}$$

Applying Lemma 2 for the first equation with $h(t) = -v(g(\boldsymbol{\phi}(t)) - q)$, we have that the solution of the dynamical system $(\Sigma)$ is

$$\psi_{\boldsymbol{\theta}(0)}(t) = \gamma_{\boldsymbol{\theta}(0)}(\underbrace{\int_0^t h(s)\mathrm{d}s}_{H(t)}) = \gamma_{\boldsymbol{\theta}(0)}(H(t))$$

Thus it holds

$$f(\psi_{\boldsymbol{\theta}(0)}(t)) = f(\gamma_{\boldsymbol{\theta}(0)}(H(t))) = A_{\boldsymbol{\theta}(0)}(H(t))$$

or equivalently

$$H(t) = A_{\boldsymbol{\theta}(0)}^{-1}(f(\psi_{\boldsymbol{\theta}(0)}(t)))$$

Plug in back to the definition of the solution, clearly we have that :

$$\psi_{\boldsymbol{\theta}(0)}(t) = \gamma_{\boldsymbol{\theta}(0)}(A_{\boldsymbol{\theta}(0)}^{-1}(f(\psi_{\boldsymbol{\theta}(0)}(t))))$$

Therefore for $X_{\boldsymbol{\theta}(0)}(f) = \gamma_{\boldsymbol{\theta}(0)} \circ A_{\boldsymbol{\theta}(0)}^{-1}(f)$, which is $C^1$ as composition of $C^1$ functions, the theorem holds.

We can perform the equivalent analysis for the $\boldsymbol{\phi}(0)$ and $g$ and prove that for each $\boldsymbol{\phi}(0)$, under the dynamics Continuous GDA (Equation 3), there is a $C^1$ function $X_{\boldsymbol{\phi}(0)} : g_{\boldsymbol{\phi}(0)} \to \mathbb{R}^n$ such that $\boldsymbol{\phi}(t) = X_{\boldsymbol{\phi}(0)}(g(t))$. $\qquad\square$

> Notice that the domains of the aforementioned functions are in fact either singleton points or open intervals. This will be important when we study the safety of initial conditions.

**Lemma 3** (Properties of $f_{\boldsymbol{\theta}(0)}$)**.** *If $\boldsymbol{\theta}(0)$ is a stationary point of $f$, then $f_{\boldsymbol{\theta}(0)}$ consists only of a single number. Otherwise, $f_{\boldsymbol{\theta}(0)}$ is an open interval.*

*Proof.* If $\boldsymbol{\theta}(0)$ is a fixed point then for the gradient ascent dynamics $\boldsymbol{\theta}(t) = \boldsymbol{\theta}(0)$ and therefore the Theorem holds trivially. On the other hand, in Theorem 1 we argued that $f(\boldsymbol{\theta}(t))$ is a continuous and strictly increasing function so it should map $(-\infty, \infty)$ to an open set and thus the theorem holds. Obviously we can prove an equivalent theorem for $g$. $\qquad\square$

Having established the informational equivalence between the parameter and functional space, we are ready to derive the induced dynamics of the distribution with which two players participate into the game.

**Lemma 4** (Restated Lemma 2)**.** *If $\boldsymbol{\theta}(t)$ and $\boldsymbol{\phi}(t)$ are solutions to Equation 3 with initial conditions $(\boldsymbol{\theta}(0), \boldsymbol{\phi}(0))$, then we have that $f(t) = f(\boldsymbol{\theta}(t))$ and $g(t) = g(\boldsymbol{\phi}(t))$ satisfy the following equations*

$$\dot{f} = -v\|\nabla f(X_{\boldsymbol{\theta}(0)}(f))\|^2(g - q)$$
$$\dot{g} = v\|\nabla g(X_{\boldsymbol{\phi}(0)}(g))\|^2(f - p)$$

*Proof.* Applying chain rule and the definition of Continuous GDA (Equation 3) we can see that :

$$\begin{cases} \dot{f} &=& \nabla f(\boldsymbol{\theta}(t))\dot{\boldsymbol{\theta}}(t) \\ \dot{g} &=& \nabla g(\boldsymbol{\phi}(t))\dot{\boldsymbol{\phi}}(t) \end{cases} \Leftrightarrow \begin{cases} \dot{f} &=& -v\|\nabla f(\boldsymbol{\theta}(t))\|_2^2 \, (g(\boldsymbol{\phi}(t)) - q) \\ \dot{g} &=& v\|\nabla g(\boldsymbol{\phi}(t))\|_2^2 \, (f(\boldsymbol{\theta}(t)) - p) \end{cases}$$

Finally using Theorem 1 we get:

$$\begin{cases} \dot{f} &=& -v\|\nabla f(X_{\boldsymbol{\theta}(0)}(f(t)))\|_2^2 & (g(\boldsymbol{\phi}(t)) - q) \\ \dot{g} &=& v\|\nabla g(X_{\boldsymbol{\phi}(0)}(g(t)))\|_2^2 & (f(\boldsymbol{\theta}(t)) - p) \end{cases}$$

$\square$

Finally, we establish that the above 2-dimensional system that couples $f, g$ together is akin to a conservative system that preserves an energy-like function. Under the safety conditions, the proposed invariant is both well-defined and equipped with interesting properties. It is easy to check that it can play the role of a pseudometric around the Nash Equilibrium of the hidden bilinear game.

**Theorem 9** (Restated Theorem 2)**.** *Let $\boldsymbol{\theta}(0)$ and $\boldsymbol{\phi}(0)$ be safe initial conditions. Then for the system of Equation 3, the following quantity is time-invariant*

$$H(f, g) = \int_p^f \frac{z - p}{\|\nabla f(X_{\boldsymbol{\theta}(0)}(z))\|^2}\mathrm{d}z + \int_q^g \frac{z - q}{\|\nabla g(X_{\boldsymbol{\phi}(0)}(z))\|^2}\mathrm{d}z$$

*Proof.* Firstly, one should notice that since $\boldsymbol{\theta}(0)$ and $\boldsymbol{\phi}(0)$ are safe initial conditions, $H(f, g)$ is well defined when $f, g$ follows the dynamics Continuous-GDA. We will examine the derivative of the proposed invariant of motion.

$$\frac{\mathrm{d}[H(f(t), g(t))]}{\mathrm{d}t} = \frac{\mathrm{d}[\int_p^{f(t)} \frac{z-p}{\|\nabla f(X_{\boldsymbol{\theta}(0)}(z))\|^2}\mathrm{d}z]}{\mathrm{d}t} + \frac{\mathrm{d}[\int_q^{g(t)} \frac{z-q}{\|\nabla g(X_{\boldsymbol{\phi}(0)}(z))\|^2}\mathrm{d}z]}{\mathrm{d}t}$$

$$= \frac{\mathrm{d}[f(t)]}{\mathrm{d}t} \times \frac{f(t) - p}{\|\nabla f(X_{\boldsymbol{\theta}(0)}(f(t)))\|^2} + \frac{\mathrm{d}[g(t)]}{\mathrm{d}t} \times \frac{g(t) - q}{\|\nabla g(X_{\boldsymbol{\phi}(0)}(g(t)))\|^2}$$

Using Theorem 4, we get

$$\frac{\mathrm{d}[H(f(t), g(t))]}{\mathrm{d}t} = -v\|\nabla f(X_{\boldsymbol{\theta}(0)}(f(t)))\|_2^2 \, (g(\boldsymbol{\phi}(t)) - q) \times \frac{f(t) - p}{\|\nabla f(X_{\boldsymbol{\theta}(0)}(f(t)))\|^2} +$$

$$v\|\nabla g(X_{\boldsymbol{\phi}(0)}(g(t)))\|_2^2 \, (f(\boldsymbol{\theta}(t)) - p) \times \frac{g(t) - q}{\|\nabla g(X_{\boldsymbol{\phi}(0)}(g(t)))\|^2}$$

$$= -v(f(t) - p)(g(t) - q) + v(f(t) - p)(g(t) - q) = 0$$

$\square$

Using the existence of the invariant function for the safe initial conditions, we will prove that the trajectory of the planar dynamical system stays bounded away from all possible fixed points. Therefore the limit behavior must be a cycle. We can also prove that the system does not just converge to a periodic orbit but it actually lies on the periodic trajectory from the very beginning. The key intuition that allows us to do this is that the level sets of $H$ are one-dimensional manifolds. To get convergence to a periodic orbit, one would require two orbits (the initial trajectory and the periodic orbit) to merge into the same one dimensional manifold, but this is not possible (requires that no transient part exists).

**Theorem 10** (Restated Theorem 3). *Let $\boldsymbol{\theta}(0)$ and $\boldsymbol{\phi}(0)$ be safe initial conditions. Then for the system of Equation 3, the orbit $(\boldsymbol{\theta}(t), \boldsymbol{\phi}(t))$ is periodic.*

*Proof.* If $(\boldsymbol{\theta}(0), \boldsymbol{\phi}(0))$ is a fixed point then it is trivially a periodic point. Suppose $(\boldsymbol{\theta}(0), \boldsymbol{\phi}(0))$ is not a fixed point, then either $f \neq p$ or $g \neq q$ (or both). Given that $H$ is invariant, the trajectory of the planar system stays bounded away from all equilibria. We will examine each case separately:

**Equilbria with $f = p$ and $g = q$** It is bounded away from these since $H(p, q) = 0$ and $H(f(\boldsymbol{\theta}(0)), g(\boldsymbol{\phi}(0))) > 0$.

**Equilibria with $f = p$ and $\nabla f = 0$** These equilibria are not achievable since they are not allowed by the safety conditions. $\nabla f = 0$ when $f = p$ means that $p$ is one of the endpoints of $f_{\boldsymbol{\theta}(0)}$. But by Lemma 3, $f_{\boldsymbol{\theta}(0)}$ is an open set and $p \in f_{\boldsymbol{\theta}(0)}$ which leads to a contradiction.

**Equilibria with $g = q$ and $\nabla g = 0$** They are also not feasible due to the safety assumption.

**Equilibria with $\nabla f = 0$ and $\nabla g = 0$** Observe that such points lie in the corners of $f_{\boldsymbol{\theta}(0)} \times g_{\boldsymbol{\phi}(0)}$. These points correspond to local maxima of the invariant function. We will prove this for one of the corners and the same proof works for all others in the same way. Let $(p^*, q^*)$ be one such corner with both $p^* > p$ and $q^* > q$. Let us take any other point $(r, z)$ with $p^* \geq r > p$ and $q^* \geq z > q$ but different from $(p, q)$. Without loss of generality let us assume $p^* > r$. Then in this region $H$ is increasing in both $f$ and $g$. Thus

$$H(r, z) < H(p^*, z) \leq H(p^*, q^*)$$

So this corner (and all the other three corners) are local maxima. A continuous trajectory cannot reach these isolated local maxima while maintaining $H$ invariant.

Thus we can create a trapping/invariant region $C$ so that $f$ and $g$ always stay in $C$ and $C$ does not contain any fixed points. By the Poincaré-Bendixson theorem, the $\alpha, \omega$-limit set of the trajectory is a periodic orbit. Thus they are isomorphic to $S^1$.

Since the gradient of $H$ is only equal to 0 at $(p, q)$

$$\nabla H = \left( \frac{f - p}{\|\nabla f(X_{\boldsymbol{\theta}(0)}(f))\|^2}, \frac{g - q}{\|\nabla g(X_{\boldsymbol{\phi}(0)}(g))\|^2} \right)$$

Therefore $H(f(\boldsymbol{\theta}(0)), g(\boldsymbol{\phi}(0))) > H(p, q)$ is a regular value of $H$. By the regular value theorem the following set is a one dimensional manifold

$$\{(f, g) \in f_{\boldsymbol{\theta}(0)} \times g_{\boldsymbol{\phi}(0)} : H(f, g) = H(f(\boldsymbol{\theta}(0)), g(\boldsymbol{\phi}(0)))\}$$

Notice that by the invariance of $H$ and definition of $\alpha, \omega-$limit sets of $(f(\boldsymbol{\theta}(0)), g(\boldsymbol{\phi}(0)))$, we know that both the trajectory starting at $(\boldsymbol{\theta}(0), \boldsymbol{\phi}(0))$, along with its $\alpha, \omega-$limit sets belong to the above manifold. Thus, their union is a closed, connected $1-$manifold and thus it is isomorphic to $S^1$.

Assume that the trajectory was merely converging to the $\alpha, \omega-$limit sets. Then our one dimensional manifold is containing two connected one dimensional manifolds: the trajectory of the system as well as the $\alpha, \omega-$limit sets . But one can easily show that this would not be a one dimensional manifold, leading to a contradiction.

Up to now we have analyzed the trajectories of the planar dynamical system of $f$ and $g$. But since we have proved that there is one to one correspondence between $\boldsymbol{\theta}$ and $f$ and $\boldsymbol{\phi}$ and $g$, the periodicity claims transfer to $\boldsymbol{\theta}(t)$ and $\boldsymbol{\phi}(t)$. $\square$

Figure 1: By the Poincaré-Bendixson theorem we know that both the $\alpha$ and the $\omega$ limit-sets are isomorphic to $S^1$. The trajectory connecting them makes the union of all three parts is not a one dimensional manifold. But by the regular value theorem on $H$, the union of all three parts is also a one dimensional manifold.

On a positive note, one can prove that the time average of $f$ and $g$ do converge as well as the utilities of the generator and discriminator.

**Theorem 11** (Restated Theorem 4). *Let $\boldsymbol{\theta}(0)$ and $\boldsymbol{\phi}(0)$ be safe initial conditions and $(\boldsymbol{P}, \boldsymbol{Q}) = \left( \binom{p}{1-p}, \binom{q}{1-q} \right)$, then for the system of Equation 3*

$$\lim_{T \to \infty} \frac{\int_0^T f(\boldsymbol{\theta}(t)) \mathrm{d}t}{T} = p, \quad \lim_{T \to \infty} \frac{\int_0^T r(\boldsymbol{\theta}(t), \boldsymbol{\phi}(t)) \mathrm{d}t}{T} = \boldsymbol{P}^\top U \boldsymbol{Q}, \quad \lim_{T \to \infty} \frac{\int_0^T g(\boldsymbol{\phi}(t)) \mathrm{d}t}{T} = q$$

*Proof.* In Theorem Theorem 3 we have discussed that the safety of the initial conditions guarantees that stationary points of $f$ and $g$ are going to be avoided. So using Lemma 2, we can integrate the following quantities over a time interval $[0, T]$ and divide by $T$.

$$\frac{1}{T} \int_0^T \frac{1}{v \|\nabla f(X_{\boldsymbol{\theta}(0)}(f(t)))\|^2} \frac{\mathrm{d}f}{\mathrm{d}t} \mathrm{d}t = -\frac{1}{T} \int_0^T (g(\boldsymbol{\phi}(t)) - q) \, \mathrm{d}t$$

$$\frac{1}{T} \int_0^T \frac{1}{v \|\nabla g(X_{\boldsymbol{\phi}(0)}(g(t)))\|^2} \frac{\mathrm{d}g}{\mathrm{d}t} \mathrm{d}t = \frac{1}{T} \int_0^T (f(\boldsymbol{\theta}(t)) - p) \, \mathrm{d}t$$

Let us define the follwoing functions of $f$ and $g$:

$$\mathbb{F}(f(t)) = v \|\nabla f(X_{\boldsymbol{\theta}(0)}(f(t)))\|^2$$

$$\mathbb{G}(g(t)) = v \|\nabla g(X_{\boldsymbol{\phi}(0)}(g(t)))\|^2$$

Thus the above dynamical system is equivalent with:

$$\frac{1}{T} \int_0^T \frac{1}{\mathbb{F}(f(t))} \frac{\mathrm{d}f}{\mathrm{d}t} \mathrm{d}t = -\frac{1}{T} \int_0^T (g(\boldsymbol{\phi}(t)) - q) \, \mathrm{d}t$$

$$\frac{1}{T} \int_0^T \frac{1}{\mathbb{G}(g(t))} \frac{\mathrm{d}g}{\mathrm{d}t} \mathrm{d}t = \frac{1}{T} \int_0^T (f(\boldsymbol{\theta}(t)) - p) \, \mathrm{d}t$$

However, by a simple change of variables we have that :

$$\int_0^T \frac{1}{\mathbb{F}(f)} \frac{df}{dt} dt = \int_{f(0)}^{f(T)} \frac{1}{\mathbb{F}(f)} df$$

$$\int_0^T \frac{1}{\mathbb{G}(g)} \frac{dg}{dt} dt = \int_{g(0)}^{g(T)} \frac{1}{\mathbb{G}(g)} dg$$

However we know that $f(t), g(t)$ for our dynamical system are periodic and bounded away from the roots of $\mathbb{F}(f), \mathbb{G}(g)$. So their integrals over a single period of $f$ and $g$ are bounded and we have that

$$\lim_{T \to \infty} \frac{1}{T} \int_0^T \frac{1}{\mathbb{F}(f)} \frac{df}{dt} dt = \lim_{T \to \infty} \frac{1}{T} \int_{f(0)}^{f(T)} \frac{1}{\mathbb{F}(f)} df = 0$$

$$\lim_{T \to \infty} \frac{1}{T} \int_0^T \frac{1}{\mathbb{G}(g)} \frac{dg}{dt} dt = \lim_{T \to \infty} \frac{1}{T} \int_{g(0)}^{g(T)} \frac{1}{\mathbb{G}(g)} dg = 0$$

Therefore,

$$\lim_{T \to \infty} \frac{1}{T} \int_0^T (g(\boldsymbol{\phi}(t)) - q)) \, \mathrm{d}t = 0$$

$$\lim_{T \to \infty} \frac{1}{T} \int_0^T (f(\boldsymbol{\theta}(t)) - p)) \, \mathrm{d}t = 0$$

which implies

$$\lim_{T\to\infty}\frac{\int_0^T g(\boldsymbol{\phi}(t))dt}{T}=q \quad \lim_{T\to\infty}\frac{\int_0^T f(\boldsymbol{\theta}(t))dt}{T}=p$$

Next, we will proceed with the argument about the time average of the objective function.

**Fact 1.** *If* $(\boldsymbol{P},\boldsymbol{Q})$ *is fully mixed Nash Equilibrium, then it holds*

$$\boldsymbol{P}^\top U\boldsymbol{G}(\boldsymbol{\phi}(t))=\boldsymbol{F}(\boldsymbol{\theta}(t))^\top U\boldsymbol{Q}=\boldsymbol{P}^\top U\boldsymbol{Q}$$
$$(\boldsymbol{F}(\boldsymbol{\theta}(t))-\boldsymbol{P})^\top U(\boldsymbol{G}(\boldsymbol{\phi}(t))-\boldsymbol{Q})=\boldsymbol{F}(\boldsymbol{\theta}(t))^\top U\boldsymbol{G}(\boldsymbol{\phi}(t))-\boldsymbol{P}^\top U\boldsymbol{Q}$$

*Proof.* It suffices to prove the first part of the claim, since the second part is its immediate consequence. Since we have conditioned that $(\boldsymbol{P},\boldsymbol{Q})$ is a fully mixed Nash Equilibrium, it holds :

$$\boldsymbol{P}^\top U\boldsymbol{Q}=\begin{pmatrix}1\\0\end{pmatrix}^\top U\boldsymbol{Q}=\begin{pmatrix}0\\1\end{pmatrix}^\top U\boldsymbol{Q}$$

Therefore:

$$\boldsymbol{F}(\boldsymbol{\theta})^\top U\boldsymbol{Q}=f(\boldsymbol{\theta})\begin{pmatrix}1\\0\end{pmatrix}^\top U\boldsymbol{Q}+(1-f(\boldsymbol{\theta}))\begin{pmatrix}0\\1\end{pmatrix}^\top U\boldsymbol{Q}=\boldsymbol{P}^\top U\boldsymbol{Q}$$

Symmetrically, it holds :

$$\boldsymbol{P}^\top U\boldsymbol{Q}=\boldsymbol{P}^\top U\begin{pmatrix}1\\0\end{pmatrix}=\boldsymbol{P}^\top U\begin{pmatrix}0\\1\end{pmatrix}.$$

Therefore

$$\boldsymbol{P}^\top U\boldsymbol{Q}=\boldsymbol{P}^\top U\begin{pmatrix}1\\0\end{pmatrix}g(\boldsymbol{\phi}(t))+\boldsymbol{P}^\top U\begin{pmatrix}0\\1\end{pmatrix}(1-g(\boldsymbol{\phi}(t)))=\boldsymbol{P}^\top U\boldsymbol{G}(\boldsymbol{\phi}(t)).$$

$\square$

Observe the following fact:

$$\frac{1}{T}\int_0^T \boldsymbol{F}(\boldsymbol{\theta}(t))^\top U\boldsymbol{G}(\boldsymbol{\phi}(t))\mathrm{d}t-\boldsymbol{P}^\top U\boldsymbol{Q}=\frac{1}{T}\int_0^T \boldsymbol{F}(\boldsymbol{\theta}(t))^\top U\boldsymbol{G}(\boldsymbol{\phi}(t))\mathrm{d}t-\frac{1}{T}\int_0^T \boldsymbol{P}^\top U\boldsymbol{Q}\mathrm{d}t$$
$$=\frac{1}{T}\int_0^T (\boldsymbol{F}(\boldsymbol{\theta}(t))-\boldsymbol{P})^\top U(\boldsymbol{G}(\boldsymbol{\phi}(t))-\boldsymbol{Q})\mathrm{d}t$$

Therefore it suffices to show that

$$\lim_{T\to\infty}\frac{1}{T}\int_0^T (\boldsymbol{F}(\boldsymbol{\theta}(t))-\boldsymbol{P})^\top U(\boldsymbol{G}(\boldsymbol{\phi}(t)-\boldsymbol{Q})\mathrm{d}t=0$$

The payoff matrix $U$ is as follows:

$$U=\begin{pmatrix}u_{0,0}&u_{1,0}\\u_{1,0}&u_{1,1}\end{pmatrix}$$

We have that

$$(\boldsymbol{F}(\boldsymbol{\theta}(t))-\boldsymbol{P})^\top U(\boldsymbol{G}(\boldsymbol{\phi}(t))-\boldsymbol{Q})=(u_{0,0}-u_{1,0}-u_{1,0}+u_{1,1})(f(\boldsymbol{\theta}(t))-p))(g(\boldsymbol{\phi}(t))-q).$$

Therefore it suffices to show that :

$$\lim_{T\to\infty}\frac{1}{T}\int_0^T (f(\boldsymbol{\theta}(t))-p))(g(\boldsymbol{\phi}(t))-q)\mathrm{d}t=0.$$

By our previous analysis in this theorem, we have already argued that

$$\lim_{T\to\infty}\frac{1}{T}\int_0^T (g(\boldsymbol{\phi}(t))-q)\mathrm{d}t=0$$

thus we only have to show that

$$\lim_{T \to \infty} \frac{1}{T} \int_0^T f(\boldsymbol{\theta}(t))(g(\boldsymbol{\phi}(t)) - q)\mathrm{d}t = 0$$

Revisiting the equations of Lemma 2:

$$\frac{f}{\mathbb{F}(f)} \frac{df}{\mathrm{d}t} = f(\boldsymbol{\theta}(t))(g(\boldsymbol{\phi}(t)) - q) \Rightarrow$$

$$\frac{1}{T} \int_0^T \frac{f}{\mathbb{F}(f)} \frac{df}{\mathrm{d}t}\mathrm{d}t = \frac{1}{T} \int_0^T f(\boldsymbol{\theta}(t))(g(\boldsymbol{\phi}(t)) - q)\mathrm{d}t$$

However using similar arguments as before we can prove that

$$\lim_{T \to \infty} \frac{1}{T} \int_0^T \frac{f}{\mathbb{F}(f)} \frac{df}{\mathrm{d}t}\mathrm{d}t = \lim_{T \to \infty} \frac{1}{T} \int_{f(0)}^{f(T)} \frac{f}{\mathbb{F}(f)}df = 0$$

implying that

$$\lim_{T \to \infty} \frac{1}{T} \int_0^T f(\boldsymbol{\theta}(t))(g(\boldsymbol{\phi}(t)) - q)\mathrm{d}t = 0$$

which completes the proof. □

# 3 Omitted Proofs of Section 5
# Poincaré recurrence in hidden bilinear games with more strategies

**Lemma 5** (Restated Lemma 3). *If $\boldsymbol{\theta}(t)$ and $\boldsymbol{\phi}(t)$ are solutions to Equation 7 with initial conditions $(\boldsymbol{\theta}(0), \boldsymbol{\phi}(0), \lambda(0), \mu(0))$, then we have that $f_i(t) = f_i(\boldsymbol{\theta}_i(t))$ and $g_j(t) = g_j(\boldsymbol{\phi}_j(t))$ satisfy the following equations*

$$\dot{f}_i = -\|\nabla f_i(X_{\boldsymbol{\theta}_i(0)}(f_i))\|^2 \left( \sum_{j=1}^{M} u_{i,j} g_j + \lambda \right)$$

$$\dot{g}_j = \|\nabla g_j(X_{\boldsymbol{\phi}_j(0)}(g_j))\|^2 \left( \sum_{i=1}^{N} u_{i,j} f_i + \mu \right)$$

*Proof.* Applying chain rule we can see that :

$$\forall i \in [N] : \dot{f}_i = \nabla f_i(\boldsymbol{\theta}_i(t))\dot{\boldsymbol{\theta}}_i(t)$$

$$\forall j \in [M] : \dot{g}_j = \nabla g_j(\boldsymbol{\phi}_j(t))\dot{\boldsymbol{\phi}}_j(t)$$

Then by the dynamics of Continuous GDA (Equation 3)

$$\forall i \in [N] : \dot{f}_i = \nabla f_i(\boldsymbol{\theta}_i(t)) \left( -\nabla f_i(\boldsymbol{\theta}_i) \left( \sum_{j=1}^{M} u_{i,j} g_j(\boldsymbol{\phi}_j) + \lambda \right) \right)$$

$$\forall j \in [M] : \dot{g}_j = \nabla g_j(\boldsymbol{\phi}_j(t)) \left( \nabla g_j(\boldsymbol{\phi}_j) \left( \sum_{i=1}^{N} u_{i,j} f_i(\boldsymbol{\theta}_i) + \mu \right) \right)$$

Clearly

$$\forall i \in [N] : \dot{f}_i = -\|\nabla f_i(\boldsymbol{\theta}_i(t))\|^2 \left( \sum_{j=1}^{M} u_{i,j} g_j(\boldsymbol{\phi}_j) + \lambda \right)$$

$$\forall j \in [M] : \dot{g}_j = \|\nabla g_j(\boldsymbol{\phi}_j(t))\|^2 \left( \sum_{i=1}^{N} u_{i,j} f_i(\boldsymbol{\theta}_i) + \mu \right)$$

Finally using Theorem 1 we know that there exist $N + M$ functions such that :

$$\boldsymbol{\theta}_i(t) = X_{\boldsymbol{\theta}_i(0)}(f_i(t))$$

$$\boldsymbol{\phi}_j(t) = X_{\boldsymbol{\phi}_j(0)}(g_j(t))$$

Combining the last two expressions we get the desired claim. $\square$

**Theorem 12** (Restated Theorem 5). *Assume that $(\boldsymbol{\theta}(0), \boldsymbol{\phi}(0), \lambda(0), \mu(0))$ is a safe initialization. Then there exist $\lambda_*$ and $\mu_*$ such that the following quantity is time invariant:*

$$H(\boldsymbol{F}, \boldsymbol{G}, \lambda, \mu) = \sum_{i=1}^{N} \int_{p_i}^{f_i} \frac{z - p_i}{\|\nabla f_i(X_{\boldsymbol{\theta}_i(0)}(z))\|^2} \mathrm{d}z + \sum_{j=1}^{M} \int_{q_j}^{g_j} \frac{z - q_j}{\|\nabla g_j(X_{\boldsymbol{\phi}_j(0)}(z))\|^2} \mathrm{d}z +$$

$$\int_{\lambda^*}^{\lambda} (z - \lambda^*) \, \mathrm{d}z + \int_{\mu^*}^{\mu} (z - \mu^*) \, \mathrm{d}z$$

*Proof.* We know that $(\boldsymbol{p}, \boldsymbol{q})$ is an equilibrium of the hidden bilinear game

$$\min_{\boldsymbol{x} \in \Delta_N} \max_{\boldsymbol{y} \in \Delta_M} \boldsymbol{x}^\top U \boldsymbol{y} \tag{1}$$

Let us make the same Lagrangian transformation we did in Section 5.

$$\min_{\boldsymbol{x}\geq 0,\mu\in\mathbb{R}}\max_{\boldsymbol{y}\geq 0,\lambda\in\mathbb{R}}\boldsymbol{x}^\top U\boldsymbol{y} + \mu\left(\sum_{i=1}^{M} y_i\right) + \lambda\left(\sum_{j=1}^{N} x_j\right) \tag{2}$$

Since $(\boldsymbol{p},\boldsymbol{q})$ is an equilibrium of the problem of Equation 1, the KKT conditions on the Problem of Equation 2 imply that there are (unique) $\lambda^*, \mu^*$

$$\forall j \in [M]: \quad \sum_{i\in[N]} u_{i,j}p_i + \mu^* = 0$$

$$\forall i \in [N]: \quad \sum_{j\in[M]} u_{i,j}q_j + \lambda^* = 0$$

We will analyze the time derivative of $H(\boldsymbol{F}(t),\boldsymbol{G}(t),\lambda(t),\mu(t))$ over the trajectory of CGDA (Equation 7).

$$H(\boldsymbol{F},\boldsymbol{G},\lambda,\mu) = \sum_{i=1}^{N}\int_{p_i}^{f_i}\frac{z-p_i}{\|\nabla f_i(X_{\boldsymbol{\theta}_i(0)}(z))\|^2}\mathrm{d}z + \sum_{j=1}^{M}\int_{q_j}^{g_j}\frac{z-q_j}{\|\nabla g_j(X_{\boldsymbol{\phi}_j(0)}(z))\|^2}\mathrm{d}z +$$

$$\int_{\lambda^*}^{\lambda}(z-\lambda^*)\,\mathrm{d}z + \int_{\mu^*}^{\mu}(z-\mu^*)\,\mathrm{d}z \Rightarrow$$

$$\frac{\mathrm{d}[H(\boldsymbol{F}(t),\boldsymbol{G}(t),\lambda(t),\mu(t))]}{\mathrm{d}t} = \sum_{i=1}^{N}\dot{f}_i\frac{f_i-p_i}{\|\nabla f_i(X_{\boldsymbol{\theta}_i(0)}(f_i))\|^2} + \sum_{j=1}^{M}\dot{g}_j\frac{g_j-q_j}{\|\nabla g_j(X_{\boldsymbol{\phi}_j(0)}(g_j))\|^2}$$

$$+ \dot{\lambda}(\lambda-\lambda^*) + (\mu-\mu^*)\dot{\mu}$$

$$\frac{\mathrm{d}[H(\boldsymbol{F}(t),\boldsymbol{G}(t),\lambda(t),\mu(t))]}{\mathrm{d}t} = \sum_{i=1}^{N}\left(\sum_{j=1}^{M} u_{i,j}g_j + \lambda\right)(p_i - f_i)$$

$$+ \sum_{j=1}^{M}\left(\sum_{i=1}^{N} u_{i,j}f_i + \mu\right))(g_j - q_j)$$

$$+ (\lambda-\lambda^*)\dot{\lambda} + (\mu-\mu^*)\dot{\mu}$$

Applying the KTT conditions we have

$$\sum_{j=1}^{M} u_{i,j}g_j + \lambda = \sum_{j=1}^{M} u_{i,j}(g_j - q_j) + \lambda - \lambda^*$$

$$\sum_{i=1}^{N} u_{i,j}f_i + \mu = \sum_{i=1}^{N} u_{i,j}(f_i - p_i) + \mu - \mu^*$$

We can now write down:

$$\sum_{i=1}^{N}\sum_{j=1}^{M} u_{i,j}g_j(p_i - f_i) + \lambda = \sum_{i=1}^{N}\sum_{j=1}^{M} u_{i,j}(g_j - q_j)(p_i - f_i) + (\lambda-\lambda^*)\sum_{i=1}^{N}(p_i - f_i)$$

$$\sum_{j=1}^{M}\sum_{i=1}^{N} u_{i,j}f_i(g_j - q_j) + \mu = \sum_{j=1}^{M}\sum_{i=1}^{N} u_{i,j}(f_i - p_i)(g_j - q_j) + (\mu-\mu^*)\sum_{j=1}^{M}(g_j - q_j)$$

Observe that summing the two expressions the $u_{i,j}$ terms cancel out. Thus we can write

$$\frac{\mathrm{d}[H(\boldsymbol{F}(t), \boldsymbol{G}(t), \lambda(t), \mu(t))]}{\mathrm{d}t} = (\lambda - \lambda^*) \sum_{i=1}^{N} (p_i - f_i) + +(\mu - \mu^*) \sum_{j=1}^{M} (q_j - g_j)$$
$$+ (\lambda - \lambda^*) \dot{\lambda} + (\mu - \mu^*) \dot{\mu}$$

Additionally we have that $\boldsymbol{p}$ and $\boldsymbol{q}$ are probability vectors so

$$\dot{\lambda} = \sum_{i=1}^{N} f_i - 1 = \sum_{i=1}^{N} (f_i - p_i)$$

$$\dot{\mu} = - \left( \sum_{j=1}^{M} g_j - 1 \right) = - \sum_{j=1}^{M} (g_j - q_j)$$

Thus

$$\frac{\mathrm{d}[H(\boldsymbol{F}(t), \boldsymbol{G}(t), \lambda(t), \mu(t))]}{\mathrm{d}t} = 0$$

$\square$

> Since the proof of the following Theorem is fairly complicated, we will firstly outline the basic steps below:
>
> 1. We first show that there is topological conjugate dynamical system whose dynamics are *incompressible* i.e. the volume of a set of initial conditions remains invariant as the dynamics evolve over time. By Theorem 4, if every solution remains in a bounded space for all $t \geq 0$, incompressibility implies recurrence.
>
> 2. To establish boundedness in these dynamics, we exploit the aforementioned invariant function.

**Theorem 13** (Restated Theorem 6). *Assume that $(\boldsymbol{\theta}(0), \boldsymbol{\phi}(0), \lambda(0), \mu(0))$ is a safe initialization. Then the trajectory under the dynamics of Equation 7 is diffeomoprphic to one trajectory of a Poincaré recurrent flow.*

*Proof.* Let us start with the dynamics of Equation 7. We we call its flow $\Phi_{\text{original}}$:

$$\Sigma_{\text{original}} : \begin{cases} \dot{\boldsymbol{\theta}}_i = -\nabla f_i(\boldsymbol{\theta}_i) \left( \sum_{j=1}^{M} u_{i,j} g_j(\boldsymbol{\phi}_j) + \lambda \right) & \dot{\boldsymbol{\phi}}_j = \nabla g_j(\boldsymbol{\phi}_j) \left( \sum_{i=1}^{N} u_{i,j} f_i(\boldsymbol{\theta}_i) + \mu \right) \\ \dot{\mu} = - \left( \sum_{j=1}^{M} g_j(\boldsymbol{\phi}_j) - 1 \right) & \dot{\lambda} = \left( \sum_{i=1}^{N} f_i(\boldsymbol{\theta}_i) - 1 \right) \end{cases}$$

In the previous theorems we have proved that $(X_{\boldsymbol{\theta}_i(0)}, X_{\boldsymbol{\phi}_j(0)})$ are diffeomorphisms. We also know that by definition we have that

$$(X_{\boldsymbol{\theta}_i(0)})^{-1}(\boldsymbol{\theta}_i) = f_i(\boldsymbol{\theta}_i) \quad \forall i \in [N]$$
$$(X_{\boldsymbol{\phi}_j(0)})^{-1}(\boldsymbol{\phi}_j) = g_j(\boldsymbol{\theta}_i) \quad \forall j \in [M]$$

We can thus define the following diffeomorphism

$$\nu : \begin{cases} f_i = (X_{\boldsymbol{\theta}_i(0)})^{-1}(\boldsymbol{\theta}_i) & \forall i \in [N] \\ b_j = (X_{\boldsymbol{\phi}_j(0)})^{-1}(\boldsymbol{\phi}_j) & \forall j \in [M] \\ \mu = \mu \\ \lambda = \lambda \end{cases}$$

Applying the transform we get a new dynamical system, whose flow we will call $\Phi_{\text{distributional}}$:

$$\Sigma_{\text{distributional}} : \begin{cases} \dot{f}_i &=& -\|\nabla f_i(X_{\boldsymbol{\theta}_i(0)}(f_i))\|^2 \left(\sum_{j=1}^M u_{i,j} g_j + \lambda\right) \\ \dot{g}_j &=& \|\nabla g_j(X_{\boldsymbol{\phi}_j(0)}(g_j))\|^2 \left(\sum_{i=1}^N u_{i,j} f_i + \mu\right) \\ \dot{\mu} &=& -\left(\sum_{j=1}^M g_j - 1\right) \\ \dot{\lambda} &=& \left(\sum_{i=1}^N f_i - 1\right) \end{cases}$$

Although $\Phi_{\text{distributional}}$ could be well defined for a wider set of points, we will focus our attention on the following set of points

$$V = f_{1_{\boldsymbol{\theta}_1(0)}} \times \cdots \times f_{N_{\boldsymbol{\theta}_N(0)}}$$
$$\times g_{1_{\boldsymbol{\phi}_1(0)}} \times \cdots \times g_{M_{\boldsymbol{\phi}_M(0)}}$$
$$\times (-\infty, \infty) \times (-\infty, \infty)$$

Observe that this choice is not problematic since:

**Claim 1.** *$V$ is an invariant set of $\Phi_{\text{distributional}}$*

*Proof.* Let
$$\boldsymbol{\mathfrak{D}}(t) = (f_1(t), \cdots, f_N(t), g_1(t), \cdots, g_M(t))$$
be the profile of all mixed strategies of all agents. Assume that there is a $t_{\text{critical}} \in \mathbb{R}$ such that starting from $\boldsymbol{\mathfrak{D}}_0$, it holds that for some $i \in [N]$, it holds that $f_i$ crosses the boundary of $V$ at time $t_{\text{critical}}$. Let us call the crossing point $\boldsymbol{\mathfrak{D}}_{\text{critical}}$. Since $f_i(t_{\text{critical}})$ is an end-point of $f_{i_{\boldsymbol{\theta}_i(0)}}$ we have that

$$\nabla f_i(X_{\boldsymbol{\theta}_i(0)}(f_i(t_{\text{critical}}))) = 0$$

and thus by the equations of $\dot{f}_i$, we have $\dot{f}_i = 0$. On the one hand, observe that for $\Phi_{\text{distributional}}(\boldsymbol{\mathfrak{D}}_{\text{critical}}, \cdot)$ we have that $f_i$ should be constant. On the other hand, for $\Phi_{\text{distributional}}(\boldsymbol{\mathfrak{D}}_0, \cdot)$ it is not the case since $\boldsymbol{\mathfrak{D}}_0 \in V$ and $\boldsymbol{\mathfrak{D}}_{\text{critical}}$ has an $f_i$ that is on the edge of $f_{i_{\boldsymbol{\theta}_i(0)}}$. Thus $\Phi_{\text{distributional}}(\boldsymbol{\mathfrak{D}}_0, \cdot)$ and $\Phi_{\text{distributional}}(\boldsymbol{\mathfrak{D}}_{\text{critical}}, \cdot)$ are different. This is a contradiction since $\boldsymbol{\mathfrak{D}}_{\text{critical}}$ and $\boldsymbol{\mathfrak{D}}_0$ belong to the same trajectory of the flow. The same argument applies for $g_j$.

$\square$

Clearly $\Phi_{\text{original}}(\{\boldsymbol{\theta}_i(0), \boldsymbol{\phi}_j(0), \mu(0), \lambda(0)\}, \cdot)$ and $\Phi(\{f_i(\boldsymbol{\theta}_i(0)), g_j(\boldsymbol{\phi}_j(0)), \mu(0), \lambda(0)\}, \cdot)$ are diffeomorphic. It thus remains to prove that $\Phi$ is Poincaré recurrent.

**Divergence Free Topological Conjugate Dynamical System** We will transform the above dynamical system to a divergence free system on different space via the following map :

$$\gamma : \begin{cases} a_i = & \mathcal{A}_i(f_i) = & \int_{p_i}^{f_i} \dfrac{1}{\|\nabla f_i(X_{\boldsymbol{\theta}_i(0)}(z))\|^2} \mathrm{d}z \quad \forall i \in [N] \\ b_j = & \mathcal{B}_j(g_j) = & \int_{q_j}^{g_i} \dfrac{1}{\|\nabla g_j(X_{\boldsymbol{\phi}_j(0)}(z))\|^2} \mathrm{d}z \quad \forall j \in [M] \\ \mu = & \mu & \\ \lambda = & \lambda & \end{cases}$$

**Claim 2.** *$\gamma$ is a diffeomorphism.*

*Proof.* Indeed,

$$\mathbb{F}_i(f) = \frac{1}{\|\nabla f_i(X_{\boldsymbol{\theta}_i(0)}(f_i))\|^2}$$

$$\mathbb{G}_j(g) = \frac{1}{\|\nabla g_j(X_{\boldsymbol{\phi}_i(0)}(g_j))\|^2}$$

are positive and smooth functions. Thus $\mathcal{A}_i(f_i), \mathcal{B}_j(g_j)$ are monotone functions and consequently bijections and are continuously differentiable. Again because of the monotonicity using Inverse Function Theorem we can show easily that $\mathcal{A}_i(f_i), \mathcal{B}_j(g_j)$ have also continuously differentiable inverse. $\square$

As a first step let us apply $\gamma$ on the equations of our dynamical system:

$$\dot{a}_i = \frac{d\mathcal{A}_i(f_i)}{df_i}\dot{f}_i = \dot{f}_i\frac{1}{\|\nabla f_i(X_{\boldsymbol{\theta}_i(0)}(f_i))\|^2} = -\left(\sum_{j=1}^{M}u_{i,j}g_j + \lambda\right)$$

$$\dot{b}_j = \frac{d\mathcal{B}_j(g_j)}{dg_j}\dot{g}_j = \dot{g}_j\frac{1}{\|\nabla g_j(X_{\boldsymbol{\phi}_j(0)}(g_j))\|^2} = \left(\sum_{i=1}^{N}u_{i,j}f_i + \mu\right)$$

Observe that on the right hand side of our equations, $f_i$ can be written as $\mathcal{A}_i^{-1}(a_i)$ and $g_j$ can be written as $\mathcal{B}_j^{-1}(g_j)$, so this is an autonomous dynamical system, whoose flow we will call $\Psi$ and whose vector field we will call $\boldsymbol{Y}$:

$$\Sigma_{\text{Preserving}} : \begin{cases} \dot{a}_i = -\left(\sum_{j=1}^{M}u_{i,j}\mathcal{B}_j^{-1}(g_j) + \lambda\right) & \dot{b}_j = \left(\sum_{i=1}^{N}u_{i,j}\mathcal{A}_i^{-1}(a_i) + \mu\right) \\ \dot{\mu} = -\left(\sum_{j=1}^{M}\mathcal{B}_j^{-1}(g_j) - 1\right) & \dot{\lambda} = \left(\sum_{i=1}^{N}\mathcal{A}_i^{-1}(a_i) - 1\right) \end{cases} \Leftrightarrow$$

$$\Sigma_{\text{Preserving}} : \begin{pmatrix} \dot{a}_i \\ \dot{b}_j \\ \dot{\mu} \\ \dot{\lambda} \end{pmatrix} = \boldsymbol{Y}(a_i, b_j, \mu, \lambda)$$

Taking the Jacobian of $\boldsymbol{Y}$, all elements across the diagonal are zero : The coordinate of $\dot{a}_i$ does not depend on $a_i$ and the same goes for all state variables. Given that the divergence of the vector field is equal to the trace of the Jacobian, we are certain that this new dynamical system is divergence free:

$$\text{div}[\boldsymbol{Y}] = 0$$

Once again we focus our attention on $\gamma(V)$ that is invariant for $\Psi$. To prove this invariant, assume that one trajectory of $\Psi$ starting from inside $\gamma(V)$ escaped it. Then given that $\gamma$ is a diffeomorphism, the corresponding trajectory of $\Phi$ will start from $V$ and also escape it, which is not possible since $V$ is invariant for $\Phi$.

**Boundness of Trajectories**   In the next section of the proof, we will show that the trajectories of $\Psi$ are also bounded. Our analysis will be based on the invariant function of Theorem 5. Note that based on the way we proved Theorem 5, the invariant supplied there is binding for **all initializations** in $V$ and not just the trajectory of $\Phi(\{f_i(\boldsymbol{\theta}_i(0)), g_j(\boldsymbol{\phi}_j(0)), \mu(0), \lambda(0)\}, \cdot)$.

We will split our proof in two cases.

**Claim 3.** *For all initializations in $\gamma(V)$, it holds that $\lambda(t), \mu(t)$ are bounded.*

*Proof.* Observe the following fact

$$\lambda(t) \to \pm\infty \Rightarrow \int_{\lambda^*}^{\lambda(t)}(z - \lambda^*)dz \to \infty \Rightarrow H \to \infty$$

The last step of this analysis comes from the fact that $H$ is a sum of non-negative terms so if one of them goes to infinity the whole sum becomes unbounded. Since initializations in $V$ start with finite values of $H$, it is necessary that $\lambda$ remains bounded. Obviously, the same proof strategy applies to the case of $\mu(t)$. $\square$

Now let us analyze the rest of the variables

**Claim 4.** *For all initializations in $\gamma(V)$, it holds that $a_i(t), b_j(t)$ are bounded.*

*Proof.* By definition

$$a_i(t) \to \pm\infty \Rightarrow \int_{p_i}^{f_i(t)}\frac{1}{\|\nabla f_i(X_{\boldsymbol{\theta}_i(0)}(z))\|^2} \to \pm\infty$$

Observe also that

$$\int_{p_i}^{f_i(t)} \frac{1}{\|\nabla f_i(X_{\boldsymbol{\theta}_i(0)}(z))\|^2} \to \pm\infty \Rightarrow \int_{p_i}^{f_i(t)} \frac{z-p_i}{\|\nabla f_i(X_{\boldsymbol{\theta}_i(0)}(z))\|^2} \to \infty$$

This is true because $z - p_i$ is bounded away from zero when $f_i$ is converging to the edges of $f_{i_{\boldsymbol{\theta}_i(0)}}$ as $p_i$ is in the interior of the set for safe initializations. Thereofe we can once again conclude that

$$a_i(t) \to \pm\infty \to H \to \infty$$

Once again for initializations in $V$, $H$ remains constant and finite. Therefore $a_i$ should be bounded. The same analysis works for $b_j$. $\square$

**Application of Poincaré Recurrence Theorem** To summarize the properties that we have established until now , we have shown that system of $\Psi$ is divergence free and has only bounded orbits. Liouville's formula also yields that $\Psi$ is a volume preserving flow. By applying Poincaré Recurrence Theorem ( Theorem 4 ) almost all initial conditions in $\gamma(V)$ of $\Psi$ are recurrent. Thus the set $W$ of all non-recurrent points in $\Psi$ has measure zero.

Using the properties of diffeomorphism, we can to propagate the recurrence behavior of $\Psi$ back to $\Phi_{\text{disitributional}}$ using Lemma 1 Thus the set of recurrent points of $\Phi$ is $\gamma^{-1}(W)$. Since diffeomorphisms preserve measure zero sets and $W$ has measure zero, the set of recurrent points of $\Phi$ has measure zero, indicating that $\Phi$ is indeed recurrent. $\square$

**Theorem 14** (Restated Theorem 7). *Let $f_i$ and $g_j$ be sigmoid functions. Then the flow of Equation 7 is Poincaré recurrent. The same holds for all functions $f_i$ and $g_j$ that are one to one functions and for which all initializations are safe.*

*Proof.* One can notice that since $f_i$ and $g_j$ are invertible functions $X_{\theta_i(0)}(\cdot)$ is totally independent of the choice $\theta_i(0)$. In other words we can substitute

$$X_{\theta_i(0)}(\cdot) = f_i^{-1}(\cdot)$$
$$X_{\phi_j(0)}(\cdot) = g_j^{-1}(\cdot)$$

Thus, in contrast to the previous theorem (Theorem 6), the construction of $\Phi_{\text{distributional}}$ does not depend on the initialization. There is a unique $\Phi_{\text{distributional}}$ for all initializations. In fact using the same map $\nu$ as in the previous theorem, we can prove that $\Phi_{\text{original}}$ is diffeomorphic to $\Phi_{\text{distributional}}$. However, using the previous theorem the flow $\Phi_{\text{distributional}}$ is Poincaré recurrent. Repeating the topological conjugacy argument of the previous theorem we can transfer the Poincaré recurrence property from the dynamical system of $\Phi_{\text{distributional}}$ to the dynamical system of $\Phi_{\text{original}}$. $\square$

## 4 Omitted Proofs of Section 6
## Spurious equilibria

**Theorem 15** (Restated Theorem 8). *One can construct functions $f$ and $g$ for the system of Equation 3 so that for a positive measure set of initial conditions the trajectories converge to fixed points that do not correspond to equilibria of the hidden game.*

*Proof.* Our strategy is to analyze the structure of the Jacobian of the vector field of Equation 3 at stationary points of $f$ and $g$. Let us call $\boldsymbol{Y}(\boldsymbol{\theta}, \boldsymbol{\phi})$ the vector field of Equation 3. Now we can write down its Jacobian

$$\mathrm{D}\boldsymbol{Y}(\boldsymbol{\theta}, \boldsymbol{\phi}) = \begin{pmatrix} -v\,(g(\boldsymbol{\phi}) - q)\,\nabla^2 f(\boldsymbol{\theta}) & -v\nabla f(\boldsymbol{\theta}) \otimes \nabla g(\boldsymbol{\phi}) \\ v\nabla g(\boldsymbol{\phi}) \otimes \nabla f(\boldsymbol{\theta}) & v\,(f(\boldsymbol{\theta}) - p)\,\nabla^2 g(\boldsymbol{\phi}) \end{pmatrix}$$

Let us focus our attention on stationary points of $f$ and $g$. Let us call them $\boldsymbol{\theta}^*$ and $\boldsymbol{\phi}^*$

$$\mathrm{D}\boldsymbol{Y}(\boldsymbol{\theta}^*, \boldsymbol{\phi}^*) = v \begin{pmatrix} -(g(\boldsymbol{\phi}^*) - q)\,\nabla^2 f(\boldsymbol{\theta}^*) & \mathbf{0}_{n \times m} \\ \mathbf{0}_{m \times n} & (f(\boldsymbol{\theta}^*) - p)\,\nabla^2 g(\boldsymbol{\phi}^*) \end{pmatrix}$$

We want to study the cases where all eigenvalues of this matrix are negative (i.e. the fixed point is stable). Let $\lambda_i(\nabla^2 f(\boldsymbol{\theta}^*))$ be the eigenvalues of $\nabla^2 f(\boldsymbol{\theta}^*)$ and $\lambda_i(\nabla^2 g(\boldsymbol{\phi}^*))$ the corresponding eigenvalues of $\nabla^2 g(\boldsymbol{\phi}^*)$. Then we know that the eigenvalues of $\mathrm{D}\boldsymbol{Y}(\boldsymbol{\theta}^*, \boldsymbol{\phi}^*)$ are

$$-v\left(g(\boldsymbol{\phi}^*) - q\right)\lambda_i(\nabla^2 f(\boldsymbol{\theta}^*)) \qquad\qquad v\left(f(\boldsymbol{\theta}^*) - p\right)\lambda_i(\nabla^2 g(\boldsymbol{\phi}^*))$$

Here we will analyze the case of $v > 0$ (the case of $v < 0$ is completely similar). To get that all eigenvalues are negative we can simply require:

- $\nabla^2 f(\boldsymbol{\theta}^*)$ and $\nabla^2 g(\boldsymbol{\phi}^*)$ are invertible.

- $\boldsymbol{\phi}^*$ is a local minimum with $g(\boldsymbol{\phi}^*) > q$. Combined with the first condition we get that $\nabla^2 g(\boldsymbol{\phi}^*)$ is positive definite.

- $\boldsymbol{\theta}^*$ is a local minimum with $f(\boldsymbol{\theta}^*) < p$. Combined with the first condition we get that $\nabla^2 f(\boldsymbol{\theta}^*)$ is positive definite.

One can observe that the second condition allows the existence of unsafe initializations if $\boldsymbol{\phi}(0)$ is in the vicinity of $\boldsymbol{\phi}^*$.

Clearly based on Theorem 5, there is a full dimensional manifold of points that eventually converge to this fixed point. Given that the manifold has full dimension, this set of points has positive measure. Additionally, $g(\boldsymbol{\phi}^*)$ and $f(\boldsymbol{\theta}^*)$ do not take the values of the unique equilibrium of the hidden Game. □

# 5 Omitted Proofs of Section 7
# Discrete Time Gradient-Descent-Ascent

The outline of this Section is the following:

1. We first review an existing result that shows that invariants of continuous time systems that have convex level sets, even though they may not be invariants for the discrete time counterparts, they are at least non-decreasing for the discrete case.

2. We show that the invariant of Theorem 5 is convex for the case of sigmoid functions. Therefore it has convex level sets.

3. We extend the construction of Theorem 8 to discrete time systems.

**Theorem 16** (Theorem 5.3. of [BP19]). *Suppose a continuous dynamic $y(t)$ has an invariant energy $H(y)$. If $H$ is continuous with convex sublevel sets then the energy in the corresponding discrete-time dynamic obtained via Euler's method/integration is non-decreasing.*

*Proof.* Let us consider a continuous time dynamical system:

$$\frac{\mathrm{d}[\boldsymbol{y}(t)]}{\mathrm{d}t} = \boldsymbol{F}(\boldsymbol{y}(t))$$

Let $t$ denote the current time instant of a trajectory with initial conditions $\boldsymbol{y}_0$. Doing discrete time gradient-descent-ascent with with step-size $\eta$ yields an approximation of $\boldsymbol{y}_{\boldsymbol{y}_0}(t + \eta)$

$$\hat{\boldsymbol{y}_{\boldsymbol{y}_0}}^{t+\eta} = \boldsymbol{y}_{\boldsymbol{y}_0}(t) + \eta\frac{\mathrm{d}[\boldsymbol{y}(t)]}{\mathrm{d}t} \tag{3}$$

To prove our theorem it suffices to show that

$$H(\hat{\boldsymbol{y}_{\boldsymbol{y}_0}}^{t+\eta}) \geq H(\boldsymbol{y}_{\boldsymbol{y}_0}(t))$$

Suppose $H(\boldsymbol{y}_{\boldsymbol{y}_0}(t)) = c$ and without loss of generality, assume $\{\boldsymbol{y}_{\boldsymbol{y}_0} : H(\boldsymbol{y}_{\boldsymbol{y}_0}) \leq c\}$ is full-dimensional. Since $\{\boldsymbol{y}_{\boldsymbol{y}_0} : H(\boldsymbol{y}_{\boldsymbol{y}_0}) \leq c\}$ is convex, there exists a supporting hyperplane $\{\boldsymbol{y}_{\boldsymbol{y}_0} : a^\mathsf{T}\boldsymbol{y}_{\boldsymbol{y}_0} = a^\mathsf{T}\boldsymbol{y}_{\boldsymbol{y}_0}(t)\}$ such that $a^\mathsf{T}\boldsymbol{y}_{\boldsymbol{y}_0} \leq a^\mathsf{T}\boldsymbol{y}_{\boldsymbol{y}_0}(t)$ for all $\boldsymbol{y}_{\boldsymbol{y}_0} \in \{\boldsymbol{y}_{\boldsymbol{y}_0} : H(\boldsymbol{y}_{\boldsymbol{y}_0}) \leq c\}$.

Because of the invariance property of $H$ over the trajectory with it holds: $H(\boldsymbol{y}_{\boldsymbol{y}_0}(t)) = c \ \forall t \in \mathbb{R}$

Therefore,

$$
\begin{aligned}
a^{\mathsf{T}}\left(\frac{d}{dt}\boldsymbol{y}_{\boldsymbol{y}_0}(t)\right) &= a^{\mathsf{T}}\left(\lim_{s\to 0^+}\frac{\boldsymbol{y}_{\boldsymbol{y}_0}(t)-\boldsymbol{y}_{\boldsymbol{y}_0}(t-s)}{s}\right)\\
&= \left(\lim_{s\to 0^+}\frac{a^{\mathsf{T}}\boldsymbol{y}_{\boldsymbol{y}_0}(t)-a^{\mathsf{T}}\boldsymbol{y}_{\boldsymbol{y}_0}(t-s)}{s}\right)\\
&\geq \left(\lim_{s\to 0^+}\frac{a^{\mathsf{T}}\boldsymbol{y}_{\boldsymbol{y}_0}(t)-a^{\mathsf{T}}\boldsymbol{y}_{\boldsymbol{y}_0}(t)}{s}\right)=0,
\end{aligned}
$$

implying

$$
\begin{aligned}
a^{\mathsf{T}}\hat{\boldsymbol{y}_{\boldsymbol{y}_0}}^{t+\eta} &= a^{\mathsf{T}}\boldsymbol{y}_{\boldsymbol{y}_0}(t)+a^{\mathsf{T}}\left(\eta\frac{d}{dt}\boldsymbol{y}_{\boldsymbol{y}_0}(t)\right)\\
&\geq a^{\mathsf{T}}\boldsymbol{y}_{\boldsymbol{y}_0}(t).
\end{aligned}
$$

For contradiction, suppose $H(\hat{\boldsymbol{y}_{\boldsymbol{y}_0}}^{t+\eta}) < c$. By continuity of $H$, for sufficiently small $\epsilon > 0$, $\hat{\boldsymbol{y}_{\boldsymbol{y}_0}}^{t+\eta} + \epsilon a \in \{\boldsymbol{y}_{\boldsymbol{y}_0} : H(\boldsymbol{y}_{\boldsymbol{y}_0}) \leq c\}$. However,

$$
a^{\mathsf{T}}(\hat{\boldsymbol{y}_{\boldsymbol{y}_0}}^{t+\eta} + \epsilon a) \geq a^{\mathsf{T}}\boldsymbol{y}_{\boldsymbol{y}_0}(t)+\epsilon||a||_2^2 > a^{\mathsf{T}}\boldsymbol{y}_{\boldsymbol{y}_0}(t) \tag{4}
$$

contradicting that $\{\boldsymbol{y}_{\boldsymbol{y}_0} : a^{\mathsf{T}}\boldsymbol{y}_{\boldsymbol{y}_0} = a^{\mathsf{T}}\boldsymbol{y}_{\boldsymbol{y}_0}(t)\}$ is a supporting hyperplane. Thus, the statement of the theorem holds. $\qquad\square$

**Lemma 6.** *The invariant of Theorem 5 is jointly convex in $\boldsymbol{\theta}$, $\boldsymbol{\phi}$, $\lambda$ and $\mu$ when $f_i$ and $g_j$ are sigmoid functions of one variable.*

*Proof.* Since $H$ is a sum of terms each involving disjoint variables, it suffices to prove that each term is convex with respect to its own variables. This follows immediately for $\lambda$ and $\mu$. Let us take one term involving $f_i$ (the same analysis works for $g_j$ terms as well). In fact we want to prove that the following function is convex

$$
\int_{p_i}^{f(\theta_i)}\frac{z-p_i}{\|\nabla f(X_{\theta_i(0)}(z))\|^2}\mathrm{d}z
$$

where $f$ is the sigmoid function. Taking the first derivative, knowing that $f' = (1-f)f$ for sigmoid we have

$$
\frac{(f(\theta_i)-p_i)\,(1-f(\theta_i))\,f(\theta_i)}{\|\nabla f(X_{\theta_i(0)}(f(\theta_i)))\|^2}
$$

$X_{\theta_i(0)}(f(\theta_i))$ is equal to $\theta_i$ since $f$ is one-to-one. Thus we can simplify

$$
\frac{(f(\theta_i)-p_i)\,(1-f(\theta_i))\,f(\theta_i)}{\|\nabla f(\theta_i)\|^2}
$$

Once again we can use the formula for the derivative of $f$

$$
\frac{(f(\theta_i)-p_i)\,(1-f(\theta_i))\,f(\theta_i)}{((1-f(\theta_i))\,f(\theta_i))^2} = \frac{(f(\theta_i)-p_i)}{(1-f(\theta_i))\,f(\theta_i)}
$$

In order to complete the convexity analysis we must take the second derivative test.

$$
\frac{d}{d\theta_i}\frac{(f(\theta_i)-p_i)}{(1-f(\theta_i))\,f(\theta_i)} = \frac{f(\theta_i)^2-2p_if(\theta_i)+p_i}{(1-f(\theta_i))^2\,f(\theta_i)^2}(1-f(\theta_i))\,f(\theta_i) = \frac{f(\theta_i)^2-2p_if(\theta_i)+p_i}{(1-f(\theta_i))\,f(\theta_i)}
$$

The only roots of the numerator are

$$
f(\theta_i) = p_i \pm \sqrt{p_i^2 - p_i}
$$

Of course for $p_i \in (0,1)$ these roots are not real. So for all $\theta_i$, $f(\theta_i) \in (0,1)$ and the second derivative is positive. This concludes our convexity proof. $\qquad\square$

Figure 2: Hidden bilinear game with two strategies having $p = 0.7$ and $q = 0.4$. The functions $f$ and $g$ are sigmoids for each player. We observe the evolution of $f$ and $g$ as well as the invariant of Theorem 2. The trajectories are close to being periodic but $H$ has began to increase even with relatively few iterations, confirming the findings of Theorem 9.

**Theorem 17** (Restated Theorem 9). *Let $f_i$ and $g_j$ be sigmoid functions. Then for the discretized version of the system of Equation 7 and for safe intializations, function $H$ of Theorem 5 is non-decreasing.*

*Proof.* First observe that given that sigmoids are invertible functions so $X_{\theta_i(0)}(f_i)$ and $X_{\phi_j(0)}(g_j)$ are independent of the initial conditions similar to the proof of Theorem 7. Thus invariant of Theorem 5 $H$ preserved by all the trajectories of the continuous time dynamical system is common across all initializations. Using Lemma 6, $H$ is convex and therefore has convex level sets. Of course it is also continuous. Using Theorem 16 we get the requested result. $\square$

**Theorem 18** (Restated Theorem 10). *One can choose a learning rate $\alpha$ and functions $f$ and $g$ for the discretized version of the system of Equation 3 so that for a positive measure set of initial conditions the trajectories converge to fixed points that do not correspond to equilibria of the hidden game.*

*Proof.* The proof follows the same construction as in the continuous case of Theorem 8. In fact, the Jacobian of the discrete time map is

$$I_{(N+M)\times(N+M)} + \alpha \mathrm{D}\boldsymbol{Y}(\boldsymbol{\theta}, \boldsymbol{\phi})$$

where $\boldsymbol{Y}$ is the vector field of the continuous time system. We can do the same construction as in Theorem 8, to get a fixed point $(\boldsymbol{\theta}^*, \boldsymbol{\phi}^*)$ such that $\mathrm{D}\boldsymbol{Y}(\boldsymbol{\theta}, \boldsymbol{\phi})$ has only negative eigenvalues and $(f(\boldsymbol{\theta}^*), g(\boldsymbol{\phi}^*)) \neq (p, q)$. Let $\lambda_{min}$ be the smallest eigenvalue of this matrix. Choose

$$\alpha < -\frac{1}{\lambda_{min}}$$

Then the Jacobian of the discrete time map has positive eigenvalues that are less than one. Therefore the discrete time map is locally a diffeomorphism and by the Stable Manifold Theorem for discrete time maps (Theorem 6), the stable manifold is again full dimensional and therefore has positive measure. $\square$

Figure 3: Hidden bilinear game with two strategies having $p = 0.4$ and $q = 0.2$. The functions are $f(x) = 0.8 + 0.2 \cdot \sigma(x)$ and $g(y) = \sigma(y)$. There is no solution of $f(x) = p$ and therefore no initialization is safe. The dynamical system converges to an equilibrium that is not game theoretically meaningful, verifying the findings of Theorem 10.

## Footnotes

[1]Jacobian of $h$ evaluated at $\boldsymbol{p}$.