[Reviews · NeurIPS 2019]

Reviewer 1



I appreciate the author's taking time to respond to comments from my review. In particular I look forward to see similar techniques applied to multi-agent hidden network zero-sum games as per their response. ------------------------------------------------------------------------------- In this paper, the authors present a (mild) generalisation of two-player zero-sum games motivated by GANS. The action space is given by finite vector spaces, and these map to a lower dimensional space via a smooth function. Payoffs are then zero-sum in this space. Although it is already known that gradient descent is difficult for games (even zero-sum games as those used), the authors show that even in this general dynamic the situation is just as dire. For classes of unique mixed equilibria games (such as matching pennies and generalisations), no matter the parametrisation functions for both players multiple undesired scenarios are possible when both players utilise a dynamical system analogue of gradient descent: -In the case that the components of the unique equilibrium are in the trajectory of the dynamical systems of each player (which already depends on the initial conditions), the orbit is either trivial (stationary), or periodic. Although this disallows convergence, the equilibrium and value of the underlying zero-sum game can be recovered as time averages. -There are cases where the equilibrium does not lie on the trajectory of the dynamical system for a given initial configuration and furthermore, in these scenarios, the dynamical system can have spurious stationary points that do not correspond to equilibria in the original game. -In higher dimensions analogous results hold, where even dynamical systems with the desired equilibrium in their trajectory are either trivial or diffeomorphic to a Poincare recurrent flow. For sigmoid functions in the hidden layer the dynamical system is indeed Poincare recurrent. -As in the two-strategy case, examples can be constructed where spurious stationary points occur as well in the gradient descent dynamic. -Finally, the authors demonstrate that the situation is even worse for discrete time approximations to the dynamical system. Overall the submission is well-written and well motivated.

Reviewer 2



I do believe the paper makes a strong contribution to the understanding of gradient dynamics in games. The proof technique looks original to me and the work is of good quality. I have checked in detail the proof of the two action case and went through the N-action case. The renaming of the theorems in appendix is very confusing but could be fixed if the author would consider that part as an appendix and not as an extended version of the full paper.

Reviewer 3



Response to the rebuttal: The rebuttal has clarified the point that this paper focuses on a (rather strict) subset of properties of GANs as the motivation. I have not objection if the introduction can be improved to highlight this, which the authors already promised to do so in the rebuttal. On the other hand, it is rather clear to me that this paper is of more direct relevance to GANs beyond the motivation presented in the paper. I wish the authors can further pursue it, at least in future work. ============================================================= The mediocre overall score I gave does not truly reflect the quality of this paper, which I believe to be well above the average of NeurIPS submissions. I gave the score due to a major concern on the motivation (detailed below), and I'm eager to increase it provided the authors can convince me in the rebuttal that the issue can be resolved. The authors claimed that the hidden bi-linear game model (2) was motivated by studying the optimization of GANs. However, it seems to me that (2) bears little connection to GANs. This is not only due to the simplified linear structure considered in the paper, but also the fact that the authors focused exclusively on "games" in the classical sense, meaning both players always have to output mixed strategies. On the other hand, none of the existing GANs possesses such a structure; rather, most of them are constrained min-max optimization problems where the outputs of both parties do not have interpretations in the mixed Nash equilibrium sense. As such, I found all results in the paper to have little implications about the training of existing GANs, which the authors claimed to have shed light on. I would like to suggest two potential approaches to fix the issue. 1. The authors can make this a pure game-theoretic paper without resorting to GANs. In order to do so, the authors need to come up with explicit examples of games where the hidden bi-linear game model is meaningful (e.g., as a simplified model for complicated agent behaviours and such). 2. Making tighter connections to GANs. For instance, I came out with the following two ways of motivating (2). (1) Learning discrete distributions by GANs. Here, N=M, meaning the outputs of both the generator and discriminator are probability distributions, where the generator aims to learn a probability distribution on N items and the discriminator tries to classify it. The matrix U represents the reward function for the discriminator, whose elements will come from (samples of) real data. The relevance to GANs then goes like "Even learning a discrete distribution by GANs through simplified bi-linear losses, the gradient descent-ascent dynamics exhibit pathological behaviour." (2) Discretization of mixed Nash Equilibrium for GANs. Take any deterministic min-max objective, which can be GANs or anything, and considered its mixed NE counterpart (see similar ideas in [1,2]). Then the optimization problem becomes \min_{\mu \in P(X)} \max_{\nu \in P(Y)} c(\mu,\nu), where P(X) and P(Y) are probability measures over the parameter spaces, and c is a bi-affine cost function. Since all the results in the paper extend to bi-affine models, we can safely assume that c is linear. Now, ***parametrize the mixed strategies*** by neural nets, which is something that people have not done but still a natural thing to do. So the outputs of the generator and discriminator are supposed to be continuous distributions which are infinite-dimensional objects, but the results in the paper show that, even if we discretize the continuous distributions into finite supports and parametrize them, the corresponding optimization problem is already very difficult, let alone in the original infinite-dimensional space. So the idea of going to mixed NE with even finite elements [2,3] results in new optimization obstacles even though it is conceptually appealing. It is perhaps worth mentioning that the above reasoning works for any nonlinear cost function c, by taking the (functional) first-order approximation which gives rise to a bi-affine problem. Minor comments and questions: 1. In definition 4, I suppose the set A needs to be of positive measure, for otherwise all measure 0 sets are Poincare recurrent. 2. In line 197, f is strictly increasing if not at stationary points. This is proved in the appendix but not mentioned in the main text which confused me for a second. 3. Although I can understand the content of definition 7, it is perhaps better to make the dependence of f and g on \lambda(0) and \mu(0) more explicit (by stating something like \nabla f_i = \nabla f_i(\theta(0), \phi(0), \lambda(0),\mu(0))). 4. I don't directly see the equivalence of (6) to hidden bi-linear games. Can the authors quickly provide the equivalence between the constrained form and the penalised form in the non-convex non-concave case? 5. In the proof, it seems that Theorem 9 only applies to constant step-size, which limits the scope of the theory. Can the authors say anything about, say, decreasing step-sizes? 6. There are spacing issues in line 29 in the appendix. 7. The transformation following line (217) in the appendix is key to the proof. Did the authors come up with such a transformation or there were similar ideas before in the literature? 8. I might have misunderstood, but in line 227 the authors stated "it is clear that \dot{a}_i is not a function of a_i", which is not very clear to me because f_i = \mathcal{A}^{-1}_i(a_i), and a_i depends on f_i. I would appreciate if the authors can explain more. [1] Y.-P. Hsieh, C. Liu, and V. Cevher, “Finding mixed nash equilibria of generative ad- versarial networks,” in 36th International Conference on Machine Learning (ICML), 2019. [2] Frans A Oliehoek, Rahul Savani, Jose Gallego, Elise van der Pol, and Roderich Groß. Beyond local nash equilibria for adversarial networks. arXiv preprint arXiv:1806.07268,2018. [3] Sanjeev Arora, Rong Ge, Yingyu Liang, Tengyu Ma, and Yi Zhang. Generalization and equilibrium in generative adversarial nets (gans). In International Conference on Machine Learning, pp. 224–232, 2017.

[Author Response · NeurIPS 2019]

We would first like to thank the reviewers for their insightful comments on our work. We appreciate that you like our paper and that you are helping us making it even stronger with your comments.

*(Motivation concerns)*. One of the questions of Reviewer #2 and #3 was about the connections of our work to training GANs using GDA dynamics. At this point, we would like to clarify the motivation behind our work. Undoubtedly, understanding GAN training dynamics in its full complexity is a very important but also difficult problem. On the other hand, as it was noted in our related work section, the recent research line in min-max optimization has mainly focused on providing guarantees on simple bilinear games (BGs). In this work, we aim to make a step towards bridging this gap. In this effort, we identify two key components of GAN training that are not captured by BGs: GAN training is *i) a non-convex non-concave minmax problem* and more importantly *ii) it involves indirect competition between the two players (networks)*. Goodfellow recognized in [4] that *ii)* is important: "In practice, however, the updates are made in parameter space, so the convexity properties that the proof relies on do not apply." These observations led us to propose the study of hidden bilinear games which combine both those properties. In order to avoid any potential source of confusion, we plan to modify the corresponding parts of abstract, introduction and conclusion sections giving more emphasis on the aforementioned properties *i)* and *ii)* than on actual GAN architectures.

*(Potential generalizations)* The majority of the questions of Reviewer #1 were centered around potential generalizations. In regards to moving to non-zero sum games where there could be more than one equilibrium, we should distinguish two elements: 1) Our setting of hidden bilinear games already allows for games with many equilibria, e.g. due to stationary points of non-convex functions $f, g$. 2) Moving completely beyond zero-sum games creates a setting that is so general that it would be hard to identify usable structure. Nevertheless, we do agree that it is very interesting to understand how much can we generalize our current setting while still salvaging interesting provable properties. Our modular setting makes it particularly amenable to such generalizations (e.g. add more than two agents that play a hidden network-zero-sum game [2], introduce different non-linearities, etc). We believe that our theoretical techniques extend to more general settings, and although these questions are beyond our current scope, we hope that our work will enable for interesting followups.

*(Technical Clarifications)* The majority of the questions that Reviewer #3 posed revolve around some technical aspects of our work. We would like to point out that even both Section 4 and 5 are framed as classical games where each player submits a distribution to the underlying BG, our results extend beyond this case. For example one can analyze Section 5 without using Lagrange multipliers yielding an unconstrained minmax problem. See also the introduction section of [3], where a similar approach was followed. Regarding the question about the equivalence of Equation 6 and Definition 1, one can check the value of the objective function in Equation 6 is finite only if the sum to one constraints are satisfied. Thus if $\theta, \phi$ has been chosen properly to satisfy them, any possible penalization by $\lambda, \mu$ can not affect the value of the game. Actually, the only difference in the non-convex non-concave case is that KKT conditions are no longer sufficient which does not affect the optimization dynamics. As far as the question about the limitation of constant step size $\eta$ in Theorem 9 is concerned, the proof of Theorem 16 that we rely on does not require $\eta$ to be the same at each iteration, as long as it remains positive. Reviewer #3 expressed also some interest in understanding the motivation behind the transformation of line 217. The major effect of this transformation is that it yields a dynamical system that is bipartite.This means that we can split the state of the system in two sets of variables so that the derivatives of the variables in one set depends only on the values of variables in the other set. Bipartite systems are known to be easier to analyze in terms of interesting properties like invariants and volume preservation [5, 1] . More explicitly, under line 225, in equation (5), RHS of the ODE describing $\dot{a}_i$ is a function that is independent of $a_i$. It is simpler to understand intuitively this observation in a standard zero-sum game such as Rock-Paper-Scissors when studied under GD. It means that the performance of the first player when playing a specific strategy e.g. Rock, does not depend on anything that he does, but on his opponents behavior.

# References

[1] J. P. Bailey and G. Piliouras. Multi-agent learning in network zero-sum games is a hamiltonian system. In *AAMAS*, 2019.

[2] Y. Cai, O. Candogan, C. Daskalakis, and C. H. Papadimitriou. Zero-sum polymatrix games: A generalization of minmax. *Math. Oper. Res.*, 2016.

[3] C. Daskalakis and I. Panageas. Last-iterate convergence: Zero-sum games and constrained min-max optimization. In *ITCS*, 2019.

[4] I. J. Goodfellow. NIPS 2016 tutorial: Generative adversarial networks. *CoRR*, abs/1701.00160, 2017.

[5] J. Hofbauer. Evolutionary dynamics for bimatrix games: A hamiltonian system? *Journal of mathematical biology*, 1996.


[Meta-Review · NeurIPS 2019]

Following the author response, consensus on acceptance was easily reached, on the grounds that the paper makes important contributions to gradient dynamics in certain zero-sum games which capture a restricted class of GANs. We encourage the authors to clarify the introduction/motivation to accurately reflect the results.